# Taxifolin Inhibits Breast Cancer Growth by Facilitating CD8+ T Cell Infiltration and Inducing a Novel Set of Genes including Potential Tumor Suppressor Genes in 1q21.3

**DOI:** 10.3390/cancers15123203

**Published:** 2023-06-15

**Authors:** Xiaozeng Lin, Ying Dong, Yan Gu, Anil Kapoor, Jingyi Peng, Yingying Su, Fengxiang Wei, Yanjun Wang, Chengzhi Yang, Armaan Gill, Sandra Vega Neira, Damu Tang

**Affiliations:** 1Department of Surgery, McMaster University, Hamilton, ON L8S 4K1, Canada; linx36@mcmaster.ca (X.L.); dongy87@mcmaster.ca (Y.D.); guy3@mcmaster.ca (Y.G.); akapoor@mcmaster.ca (A.K.); peng.jingyi96@outlook.com (J.P.); suy36@mcmaster.ca (Y.S.); gilla81@mcmaster.ca (A.G.); veganeis@mcmaster.ca (S.V.N.); 2Urological Cancer Center for Research and Innovation (UCCRI), St Joseph’s Hospital, Hamilton, ON L8N 4A6, Canada; 3The Research Institute of St Joe’s Hamilton, St Joseph’s Hospital, Hamilton, ON L8N 4A6, Canada; 4The Genetics Laboratory, Longgang District Maternity and Child Healthcare Hospital of Shenzhen City, Shenzhen 518174, China; haowei727499@163.com; 5Jilin Jianwei Songkou Biotechnology Co., Ltd., Changchun 510664, China; sengongjianwei@sina.com; 6Benda International INC., Ottawa, ON K1X 0C1, Canada; chengg_ca@yahoo.com

**Keywords:** taxifolin, breast cancer, overall survival, biomarker, immunosuppressive microenvironment

## Abstract

**Simple Summary:**

Breast cancer is the leading cause of cancer deaths in women. The current management of patients with breast cancer needs improvement in two aspects: risk estimation, which forms the basis for treatment, and effective therapies. In our research, we discovered taxifolin to inhibit breast cancer via increasing the expression of 36 genes. These genes, as a group, effectively predict the death probability in patients with breast cancer. Importantly, aggressive breast cancers are frequently associated with increases in the 1q21.3 DNA region, which is home to the *HNRN*, *KPRP*, *CRCT1*, and *FLG2* genes. These four genes can inhibit breast cancer, and their expressions are induced by taxifolin, revealing a novel mechanism by which taxifolin suppresses breast cancer. This study thus advances our ability in relation to the risk estimation and treatment of breast cancer. Through using taxifolin as a nutritional supplement, our research reveals an intriguing application of taxifolin in the clinical management of patients with breast cancer.

**Abstract:**

Taxifolin inhibits breast cancer (BC) via novel mechanisms. In a syngeneic mouse BC model, taxifolin suppressed 4T-1 cell-derived allografts. RNA-seq of 4T-1 tumors identified 36 differentially expressed genes (DEGs) upregulated by taxifolin. Among their human homologues, 19, 7, and 2 genes were downregulated in BCs, high-proliferative BCs, and BCs with high-fatality risks, respectively. Three genes were established as tumor suppressors and eight were novel to BC, including *HNRN*, *KPRP*, *CRCT1*, and *FLG2*. These four genes exhibit tumor suppressive actions and reside in 1q21.3, a locus amplified in 70% recurrent BCs, revealing a unique vulnerability of primary and recurrent BCs with 1q21.3 amplification with respect to taxifolin. Furthermore, the 36 DEGs formed a multiple gene panel (DEG36) that effectively stratified the fatality risk in luminal, HER2+, and triple-negative (TN) equivalent BCs in two large cohorts: the METABRIC and TCGA datasets. 4T-1 cells model human TNBC cells. The DEG36 most robustly predicted the poor prognosis of TNBCs and associated it with the infiltration of CD8+ T, NK, macrophages, and Th2 cells. Of note, taxifolin increased the CD8+ T cell content in 4T-1 tumors. The DEG36 is a novel and effective prognostic biomarker of BCs, particularly TNBCs, and can be used to assess the BC-associated immunosuppressive microenvironment.

## 1. Introduction

Breast cancer (BC) has overtaken lung cancer as the most frequently diagnosed cancer globally, accounting for 11.7% (2.3 million) of new cases, while in women, BC is the leading cause of mortality among all cancers [1]. The disease consists of different histological subclasses based on the expression of the estrogen receptor (ER), progesterone receptor (PR), and HER2. Moreover, tumors are classified as ER+ (70%), HER2+ (15–20%) and triple negative (TN, i.e., negative for ER, PR, and HER2; 15% of BC cases) [2]. These histological subtypes have arguably the most important impact on clinical applications. ER+ BCs have the most favorable prognosis, followed by HER2+ and TNBCs. The expression of ER and HER2 is widely used for targeted therapy [2]. In this regard, ER and HER2 possess prognostic biomarker potential and therapeutic value. TNBCs are associated with DNA repair deficiency [3]. This feature is explored for chemotherapies, including DNA-crosslink platinum-based chemotherapy and poly (ADP-ribose) polymerase (PARP) inhibitors targeting TNBCs with BRCA mutations [2,4]. Nonetheless, TNBCs are highly heterogenous and generally lack features with equivalent biomarker and therapeutic potentials compared to BC subtypes such as ER+ and HER2+. Despite the above knowledge, the current therapeutic options for recurrent and metastatic BCs are not effective. This remains the situation even with the intrinsic BC classification developed according to gene expression, although this classification enhances prognostic risk stratification. Gene profiling categorized BCs into five intrinsic subtypes: luminal A and B (ER+), HER2-enriched, basal-like, and claudin-low, with the latter two displaying TNBC features [5,6,7,8].

BC pathogenesis and progression are promoted by genome instability and inflammation, both of which are facilitated by oxidative stress [9]. Activities countering reactive oxygen species (ROS) can attenuate BC. Taxifolin (dihydroquercetin) is a flavonoid with established antioxidant and anti-inflammatory properties [10]. As a nutritional supplement, taxifolin displays multiple health-promoting effects [11,12,13]. Its anticancer activities were observed in a range of cancer cells both in vitro and in vivo (xenografts), including lung cancer [14], colorectal cancer [15,16,17], liver cancer [18], skin cancer [19], osteosarcoma [20], gastric cancer [21], glioma [22], prostate cancer [23], cervical cancer [24,25], and breast cancer [26]. Activation of the Nrf-mediated anti-ROS cytoprotective pathway, Wnt/β-catenin signaling, and p53 contributed to the taxifolin-derived anticancer activities in colon cancer, breast cancer (BC), Ewing’s sarcoma, and cervical cancer [24,27,28]. Inhibition of CYP1A1 and CYP1B1 (cytochrome P450s/CYPs) was also a mechanism of taxifolin-derived inhibition of BC [26]. Collectively, the evidence supports the existence of a general tumor-suppressive action of taxifolin toward a wide range of cancer types. Nonetheless, the impact of taxifolin on BC and the underlying mechanisms require further investigation.

We report here a systemic analysis of taxifolin-derived anti-BC activities. By investigating taxifolin’s inhibitory actions in allografts produced from 4T-1 mouse BC cells and the RNA-seq-derived gene profile, we identified *n* = 36 differentially expressed genes (DEGs) upregulated in 4T-1 tumors treated with taxifolin compared to those treated with DMSO. Their human homologue genes were largely downregulated in BC compared to normal breast tissues and contained multiple established tumor suppressors as well as genes with tumor-suppressive functions that were novel to BC. The latter included four genes residing in 1q21.3, a locus frequently amplified in recurrent BCs [29], providing a mechanistic insight regarding taxifolin’s suppression of BC. Furthermore, these 36 DEGs form an effective multigene panel (DEG36) in the stratification of survival probability in relation to all the intrinsic BC subtypes. The prediction was the most robust in the basal-like and TNBC subtypes. The DEG36 predicts the exclusion of cytotoxic lymphocytes (CD8+ and NK cells) in high-risk ER+ and TNBCs. Accordingly, 4T-1 tumors treated with taxifolin displayed a significant elevation of CD8+ T cells compared to 4T-1 tumors treated with DMSO, supporting the notion of taxifolin inhibiting BC in part via facilitating immune reactions. Collectively, this research opens doors for the potential utilization of taxifolin in BC therapy and in the mortality risk stratification of TNBCs.

## 2. Materials and Methods

### 2.1. Data Sources

The BC datasets used in this research included the METABRIC [30] and the TCGA PanCancer Atlas Breast Cancer dataset [31] organized by cBioPortal [32,33]. The Human Protein Atlas database (https://www.proteinatlas.org/) (accessed on 27 December 2022) was also utilized. Patients (*n* = 1980) in the METABRIC dataset were treated with hormone therapy for all ER+ tumors (*n* = 1216, 61%), radiotherapy (59.2%), and chemotherapy (20.8%). The TCGA cohort consisted of patients primarily treated with surgery; the other utilized treatments included neoadjuvant treatment, radiation adjuvant therapy, and pharmaceutical adjuvant therapy for 1.2%, 6.7%, and 10.5% of patients, respectively.

### 2.2. Programs and Websites

The tools utilized included the R2: Genomics Analysis and Visualization Platform (http://r2.amc.nl) (accessed on 27 October 2022), UALCAN [34], and Metascape [35]. The R *dplyr*, *survival*, *Maxstat*, *cutpointr*, and other packages were also used.

### 2.3. Profiling BC-Associated Immune Cells

Immune cell infiltration in BC was determined using RNA-seq data from the TCGA PanCancer Atlas Breast Cancer dataset with CIBERSORT [36], Epic [37], MCP [38], Quantiset [39], xCell, and ssGSEA [40] within the R *immunedeconv* and *SMDIC* packages (https://cran.r-project.org/web/packages/SMDIC/index.html; accessed on 11 November 2022). Tumor-associated CD8+ T cells were also determined using immunohistochemistry staining.

### 2.4. Model Size Optimization

The DEG36 was optimized using the sequential and golden selection (gselection) method within the R *BeSS* package (https://cran.r-project.org/web/packages/BeSS/index.html) (accessed on 26 October 2022).

### 2.5. Signature Score Assignment for Individual Tumors

The coefficient (coef) of the DEG36 component genes in predicting the overall survival probability of patients with BC was determined via multivariate Cox regression (the R *Survival* package). The risk scores for individual tumors were calculated as follows: Sum (coef_1_ × Gene_1exp_ + coef_2_ × Gene_2exp_ + … …+ coef_n_ × Gene_nexp_), where coef_1_ … coef_n_ were the coefs of individual genes and Gene_1exp_ … … Gene_nexp_ were individual gene expressions.

### 2.6. Immunohistochemistry (IHC)

IHC was performed as we have previously described [41]. Deparaffinization of slides was carried out in xylene, followed by ethanol clearance and antigen retrieval by means of heat treatment in a sodium citrate buffer (pH = 6.0). Non-specific binding sites were blocked with PBS containing 1% BSA and 10% normal goat serum (Vector Laboratories, Newark, CA, USA) for 1 h, followed by the addition of the anti-CD8 primary antibody (Cell Signaling #D4W2Z, 1:100) overnight at 4 °C and biotinylated goat anti-rabbit IgG for 1 h at room temperature. Chromogenic reaction (Vector Laboratories) and slide counterstaining with hematoxylin (Sigma Aldrich, St. Louis, MI, USA) were then carried out. Images were captured and analyzed with ImageScope software (Leica Microsystems Inc, Wetzlar, Germany); quantification was performed by counting the total number of cells and CD8+ cells; and CD8+ cells were expressed as the % of CD8+ cells. Non-specific IgG was used as a negative control. IHC images were also downloaded from the Human Protein Atlas (https://www.proteinatlas.org/) (accessed on 27 December 2022) using the HPAanalyze R package [42].

### 2.7. Culture of 4T-1 Cells, Generation of 4T-1 Tumors, and Taxifolin Treatment

Mouse 4T-1 BC cells were purchased from ATCC (American Type Culture Collection, Manassas, VA, USA; ATCC CRL-2539) and cultured in RPMI-1640 medium with 10% fetal bovine serum (FBS). Mycoplasma contamination was routinely monitored using a mycoplasma PCR detection kit (Abcam, Cambridge, UK, ab289834). Tumors were produced via subcutaneous (s.c.) implementation of 4T-1 (10^4^) cells resuspended in PBS into 6- to 8-week-old BALB/CJ mice (Jackson 000651, Winchester, VA, USA). Taxifolin was dissolved in 100 μL of PBS with 10% DMSO and administrated intraperitoneally (ip) into mice on Day 3 post tumor implantation, followed by an alternate schedule of 3 and 4 days between injections (i.e., injection twice per week) at a dose of 50 mg/kg. The tumor volume was determined following published conditions [41]. The 4T-1 cells were also treated with DMSO or taxifolin in vitro at the designed doses until colonies in the control treatment were formed. Surviving cells were stained with crystal violet (0.5%); the staining area was quantified using Image J and analyzed by means of a one-way ANOVA followed by a post-hoc (Turkey’s) test using GraphPad Prism 9.

Taxifolin (dihydroquercetin) was provided by Jilin Jianwei Songkou Biotechnology Co., Ltd., Guangzhou, China, with a purity of >90%. Taxifolin was freshly dissolved in DMSO (dimethyl sulfoxide) at 50 mg/kg/10 μL and diluted in PBS prior to animal and cell application. DMSO diluted in PBS was used as the negative control.

### 2.8. RNA Sequencing Analysis

The RNA sequencing analysis was performed following our established conditions [43]. RNA was extracted from 4T-1 tumors treated with DMSO or taxifolin (*n* = 5 per group) using an miRNeasy Mini Kit (Qiagen, No. 217004). An equal amount of RNA/sample was used for the enrichment of poly(A) mRNA using NEBNext^®^ Poly(A) mRNA Magnetic Isolation Modules. Libraries were prepared with unique dual indexes and sequenced by the McMaster Genomics Facility using a pair-end 2 × 50 bp configuration on an Illumina NextSeq 2000 P3 flow cell, with 10 M clusters aimed per sample. The RNA-seq reads were processed and analyzed with Galaxy (https://usegalaxy.org/) (accessed on 23 October 2022). Low-quality reads and adaptor sequences were removed. Alignment with the mouse genomic sequence (mm10) was carried out using HISAT2. The read counts were then performed using the “Featurecounts” function. Differential gene expression was determined using DESeq2. The enrichment analyses were carried out using Metascape [35].

### 2.9. Quantitative Real-Time PCR

RNA was isolated from the 4T-1 tumors using the Iso-RNA Lysis Reagent (5 PRIME). Reverse transcription was achieved using Superscript III (Thermo Fisher Scientific, Waltham, MA, USA). Quantitative real-time PCR was performed using the ABI 7500 Fast Real-Time PCR System (Applied Biosystems, Foster, CA, USA) with SYBR-green (Thermo Fisher Scientific). Fold changes were determined using the following formula: 2^−ΔΔCt^. The real-time PCR primers used in this research are documented in Appendix A.

### 2.10. Statistical Analysis

The Kaplan–Meier curves and log-rank test were conducted using the R *Survival* package and GraphPad Prism 7. The Cox regressions were analyzed using the R *Survival* package. ROC (receiver-operating characteristic) curves were generated using the R *PRROC* package. A multiple *t*-test with a *p* value adjusted via the Holm method (Holm–Bonferroni method) was performed using R. The other statistical analyses included the Mann–Whitney test, Chi-Square test and one-way ANOVA using GraphPad Prism 7 and SPSS 26. Data were presented as the mean ± standard deviation (SD). A value of *p* < 0.05 was considered statistically significant.

## 3. Results

### 3.1. Taxifolin Inhibits Breast Cancer Progression

Taxifolin possesses general anticancer properties [13]; however, its tumor suppressive activities have only been limitedly studied, including its actions in relation to BC [26,28]. To further investigate taxifolin’s anti-BC functions, we analyzed its impact on BC generated by murine 4T-1 cells in intact (i.e., fully immunocompetent) mice. We first confirmed the anti-proliferation potential of taxifolin in 4T-1 cells in vitro, as evidenced by the taxifolin-derived inhibition of colony formation in a dose-dependent manner (Figure 1A,B) with a calculated IC50 of 25.6 µM. We subsequently optimized taxifolin’s dose using doses of 5, 25, 50, and 100 mg/kg in the inhibition of 4T-1 tumor growth. In our system, a 50 mg/kg dosage achieved clear anticancer activity, which was not enhanced in the 100 mg/kg group. As tumors produced by 4T-1 cells show a high propensity for skin lesion development, we observed that tumors were produced with 100% efficiency at 10^4^ cells suspended in PBS. We further noticed 50 mg/kg taxifolin with IP (intraperitoneal) injection every 3 days did not affect mouse weight gain (Appendix A), which suggests the absence of overt toxicity associated with this experimental setting. With the optimized experimental conditions, we analyzed the impact of taxifolin on tumor growth in vivo. Considering variations in tumor growth and the presence of distant metastasis that affected the growth of primary tumors, we performed both median- and mean-based statistical analysis. Based on the tumor volume and number of primary tumors available, we noticed that at day 20, the median-based statistical analysis revealed a reduction in tumor growth with taxifolin treatment (Figure 1C). Similarly, the mean-based statistical analysis showed more rapid growth kinetics for tumors treated with DMSO compared to those treated with taxifolin (Figure 1D). The differences in tumor growth between the two groups at days 20, 23, and 25, which were not statistically significant (*p* > 0.05), were likely attributed to variations in tumor growth (Figure 1D). The attenuation of tumor growth by taxifolin was further supported by a significant delay in reaching the endpoint, which was defined by a decline in health and/or tumor size, in mice bearing 4T-1 tumors (Figure 1E).

To analyze the factors contributing to the taxifolin-derived inhibition of 4T-1 tumor growth, we performed RNA-seq on tumors treated with DMSO (*n* = 5) or taxifolin (*n* = 5). Taxifolin upregulated 36 differentially expressed genes (DEGs) (Figure 1F). DEGs were defined as *q* < 0.05 and a fold change ≥ 1.5. We have made efforts to confirm some DEGs in tumors treated with DMSO and taxifolin (Figure 1G, Appendix A). Enrichment analysis suggested that these DEGs facilitated skin development (GO:0043588), multicellular organism water homeostasis (GO:0050891), and formation of the cornified envelope (R-MMU-6809371) (Figure 1H). The latter two pathways are related to skin development, such as the formation of the cornified envelope. Epidermal keratinocyte cornification, a form of programmed cell death, is essential for the formation of the surface skin barrier [44]. The data indicate a novel process relevant to BC suppression.

### 3.2. The Presence of Multiple Tumor-Suppressive Components in Taxifolin-Induced DEGs

To study the relevance of the 36 DEGs to the taxifolin-derived inhibition of 4T-1 tumor growth, we first matched the murine DEGs to their 31 human homologue genes and searched for their involvement in BC using PubMed (Table 1). Genes that appeared in more than 10 articles related to BC included *n* = 9 genes: CST6, TRIM29, KRT15, KLK7, CSTA, MME, CXCL13, FABP4, and AQP3 (Table 1). CST6 [45], TRIM29 [46], and CSTA (an ER target gene) [47] possessed tumor-suppressive actions toward BC (Table 1). KRT15 [48] and KLK7 [49] were positively associated with favorable prognosis in BC patients. MME was expressed at high levels in dormant BC cells homed in the bone and lung as well as positively correlated with a favorable prognosis [50,51]. The presence of CXCL13 in the 4T-1 tumor microenvironment facilitated the immune response and passively correlated with higher overall survival (OS) in patients with BC [52,53]. The other two genes among these nine genes, FABP4 [54] and AQP3 [55], were reported to facilitate BC (Table 1).

For the remaining genes, their involvements in BC are either unknown (*n* = 8) or only limitedly studied (*n* = 13) (Table 1). In the latter group, KRT16 [60], SERPINB3 [61], and ATP6V1C2 [67] exhibited activities favoring BC, while FA2H [58] and HRNR [64] possessed tumor-suppressive activities toward BC (Table 1). ALDH3B2 [56], RDH16 [57], CD209 [62], and PSAPL1 [68] were component genes in multigene biomarkers of BC (Table 1). KRT10 [59] and CYP2F1 [63] were expressed in BC, while mutations in KPRP [65] and FLG2 [66] were detected in BC. Collectively, the evidence shows that nine genes (CST6, TRIM29, CSTA, KRT15, KLK7, MME, CXCL13, FA2H, and HRNR) negatively impact BC, while five genes (FABP4, AQP3, KRT16, SERPINB3, and ATP6V1C2) are likely to facilitate BC (Table 1). This knowledge suggests that upregulation of the 36 DEGs might be relevant to taxifolin’s inhibition of BC.

These human homologous DEG genes are not randomly distributed in the chromosome loci (Table 1). The 17.q21.2 locus and 1q21.3 locus contained three and four genes, respectively (Table 1). The four genes in 1q21.3, namely HRNR, CRCT1, KPRP, and FLG2, were amplified in two large BC datasets: TCGA and METABRIC (Figure 2A). The METABRIC cohort contained more aggressive BCs than the TCGA cohort, as evidenced by its mortality rate of 57.7% (1144/1981) compared to the mortality rate of 14% (155/1104) for the TCGA cohort (cBioPortal). Of note, amplification of these four genes occurred at a higher rate in the METABRIC cohort (Figure 2A). In comparison, amplification of the three genes, KRT10, KRT15, and KRT16, within the 17q21.2 locus occurred similarly in both cohorts and at a much lower rate (Appendix A). Furthermore, genomic alterations for all the genes exhibited no major overlap except the four genes (HRNR, CRCT1, KPRP, and FLG2) located in 1q21.3 (Appendix A). Additionally, we noticed deep deletions of the TRIM29 and FA2H genes in the TCGA cohort (Appendix A), which is consistent with their reported tumor-suppressive roles in BC (Table 1).

### 3.3. Amplification of HRNR, CRCT1, KPRP, and FLG2 Genes Does Not Lead to Their Upregulation

The impressive amplification of the HRNR, CRCT1, KPRP, and FLG2 genes implies their upregulation in BC with the increase in their gene copy. However, in both the TCGA and METABRIC cohorts, these four genes were comparably expressed in BCs with and without the amplification (Figure 2B,C). Except for HRNR, the other three genes were expressed at low levels in both cohorts (Figure 2B,C). Amplification of 1q21.3 occurred in 10–30% of primary tumors and 70% of recurrent BCs [29]. The amplification was associated with BC recurrence and poor prognosis, and genes within 1q23.1 contributed to BC progression, including S100A7, S100A8, S100A9, and IRK1 (IL-1 receptor-associated kinase 1) [29]. Amplification of 1q21.3 likely covered the intact region, as evidenced by the amplification of TUFT1 and EFNA3 bordering 1q21.3 [29]. This amplification occurred in all the major BC subtypes [29]. These observations are in line with our analyses with respect to the co-amplification of HRNR, CRCT1, KPRP, and FLG2 across all the major BC categories; however, we could not demonstrate an enrichment of either BC progression (disease-free survival/DFS) or mortality (overall survival status) in BC with the amplification (Figure 2A). Consistent with the amplification of 1q21.3 [29], we observed essentially co-amplification of our genes (HRNR, CRCT1, KPRP, and FLG2) with S100A8 in both the METABRIC and TCGA cohorts (Appendix A). In agreement with a previous publication [29], amplification of the S100A8 gene led to S100A8 upregulation in both cohorts (Appendix A).

The non-upregulation status of the HRNR, CRCT1, KPRP, and FLG2 genes in BC with their gene amplification suggests their potential actions in relation to BC suppression. Upregulation of HRNR was reported in apoptotic BC cells [64]. HRNR and FLG2 are paralog genes located in the cornified envelope and play roles in keratinocyte cornification [69], a form of programmed cell death leading to the formation of the surface skin barrier [44]. KPRP (keratinocyte proline-rich protein) may also facilitate cornification [70]. Like HRNR and FLG2, CRCT1 is a component of the epidermal differentiation complex (EDC) and likely contributes to cornification. CRCT1 was reported to display tumor-suppressive actions toward esophageal squamous cell cancer [71]. Collectively, the evidence suggests that HRNR, CRCT1, KPRP, and FLG2 promote cornification, which might underlie their potential tumor-suppressive action in relation to BC (see Discussion for details). This concept is appealing considering their induction by taxifolin (Figure 1F,G) and supported by the general low expression of the HRNR, KPRP, and FLG2 proteins in normal breast tissues and BCs (Figure 2D and Appendix A).

### 3.4. Downregulation of Taxifolin-Induced DEGs Following BC Pathogenesis

To further analyze the relevance of the DEGs to the taxifolin-derived inhibition of 4T-1 tumor growth, we studied their expression in BCs vs. normal breast tissues. Downregulation of mRNA expression in BCs occurred for AQP3, ASPRV1, ATP6V1C2, CD209, FABP4, HRNR, KLK7, KRT15, KRT16, LY6G6C, MME, and TRIM29 (*n* = 12 genes, Figure 3A). Reductions in CD209, FABP4, HRNR, KRT15, MME, and TRIM29 at the protein level in BCs compared to normal breast tissues were also demonstrated (Figure 3B). While CST6, CXCL13, KRT10, and SBSN were upregulated at the mRNA level in BCs (Figure 3A), their protein expressions were reduced (Figure 3B). We thus regard these four genes as being downregulated in BC. Additionally, CSTA and KRT79 were only downregulated at the protein level in BC (Figure 3B). Collectively, we could demonstrate the downregulation of 18 genes (12 + 4 +2) in BC compared to normal breast tissues, which constitute 58% (18/31) of the human homologues matched from the mouse DEGs (Table 1). Most downregulations are in line with either the genes’ functions in suppressing BC or their associations with a favorable prognosis (Table 1). While published evidence indicated FABP4 [54] and AQP3 [55] to facilitate BC, both were downregulated in BC (Figure 3). Furthermore, BCs with elevated FABP4 expression were at a reduced level of proliferation, as evidenced by Ki67 expression (Figure 4A). BCs with high levels of AQP3 expression correlated with better survival (Figure 4B). Our study thus suggests that the involvement of FABP4 and AQP3 in BC requires further investigation.

A reverse association with BC proliferation was also observed in ASPRV1, CD209, KLK7, KRT15, LY6G6C, and TRIM29 (Figure 4A). A high level of MME expression correlated with a low level of BC proliferation (Figure 4A) and an increase in patient survival (Figure 4B). Collectively, the above analyses support a general negative association between taxifolin-induced genes and BC.

### 3.5. Prediction of Survival Probability by DEGs Relevant to Taxifolin Treatment

BC can be categorized into the luminal A, luminal B, HER2-enriched, basal-like, claudin-low, and normal-like subtypes based on the gene expression profile. Luminal A and luminal B BCs are ER+, while basal-like, claudin-low, and normal-like tumors are TNBCs. Based on this knowledge, individual DEGs possess prognostic biomarker values in relation to predicting mortality in the intrinsic BC subtypes (Figure 5), suggesting that the expressions of these genes can be used in combination with the PAM50 classifier of the molecular BC subtypes [5] to improve risk stratification.

To further examine the biomarker potential of the taxifolin-induced DEGs, we downloaded the METABRIC dataset containing DEG expression and the relevant clinical features from cBioPortal [32,33]. Among the 36 mouse DEGs that resulted from taxifolin treatment (Figure 1F), 29 were present in the METABRIC dataset. For consistency, we named these 29 genes as the DEG36. To analyze the panel’s biomarker potential, we produced the DEG36 risk score in predicting the overall survival (OS) probability for individual tumors as ∑(coef_i_ × Gene_iexp_)_n_ (coef_i_: Cox coefficient of gene_i_, Gene_iexp_: expression of Gene_i_, *n* = 29). The coefs were derived from the multivariate Cox model. The DEG36 score effectively stratified the mortality risk of luminal A (*n* = 679), luminal B (*n* = 461), HER2+ (HER2-enriched, *n* = 202), basal-like (*n* = 199), claudin-low (*n* = 199), and normal-like (*n* = 140) BC (Figure 6A); i.e., in the METABRIC cohort (*n* = 1880), the DEG36 displayed a significant potential in separating BCs with a high fatality risk from those with a low risk. The separation was more robust in basal-like BCs compared to luminal and HER2+ BCs (Figure 6A), which is in accordance with the generation of these DEGs from the 4T-1 tumor, a mouse TNBC tumor [74]. In all the above intrinsic BC subtypes, the DEG36 score predicted the mortality risk at a hazard ratio (HR) > 2.5 (Figure 6B). Except for HER2+ BCs, the DEG36 score discriminated the fatality risk with ROC–AUC (receiver operating characteristic–area under the curve) values around 0.7 and a PR (precision-recall)–AUC around 0.8 (Figure 6C). The high PR–AUC value was in part attributed to the high rate (>50%) of mortality for patients in the METABRIC cohort. Collectively, the above analyses reveal a novel and effective potential of the DEG36 in predicting the OS probability, particularly in basal-like BCs.

The DEG36 as a panel is more powerful compared to individual genes in predicting a poor prognosis (comparing Figure 5 and Figure 6B), which was partially attributed to the minimal correlations among these DEGs in the intrinsic BC subtypes (Appendix A), suggesting their potentially different roles in impacting BC. To further explore the biomarker potential of the DEG36, we randomly divided the luminal A BCs (*n* = 679, the subgroup with the greatest patient numbers in METABRIC) into a training and testing population at a ratio of 6:4. Variables in the DEG36 were selected to predict the survival probability using the R BeSS package (Cox model-based). Both sequential and golden selection methods optimized the model size to *n* = 23 variables (DEGs) (Appendix A), which we named as the SigDEG23 (Table 1). The SigDEG23 risk score discriminated poor OS with comparable efficiency in both the training and testing subpopulations, as evidenced by their time-dependent ROC curves (Appendix A) and its ability in stratifying the fatality risk of luminal A BCs in both subpopulations (Appendix A).

In comparison to the DEG36, the SigDEG23 separated low- and high-fatality BCs with comparable effectiveness in all the intrinsic subtypes of BC except HER2+ BC (Appendix A). Both the SigDEG23 and DEG36 discriminated poor OS with comparable ROC–AUC and PR–AUC values (Figure 7A). Their risk scores were highly correlated across the molecular BC subtypes (Figure 7B). Collectively, the SigDEG23 almost fully recapitulated the biomarker potential of the DEG36, not only in luminal A tumors, in which the SigDEG23 was optimized, but also in other intrinsic subtypes of BC. Importantly, like the DEG36, the SigDEG23 was a much better biomarker for basal-like BC compared to the other BC subtypes (Appendix A). This property of the DEG36 and SigDEG23 is important given the lack of prognostic biomarkers in TNBCs, 80% of which are basal-like tumors [75].

### 3.6. Validation of the DEG36 and SigDEG23

We further validated the DEG36 and SigDEG23 using the TCGA PanCancer Atlas BC cohort, another large BC dataset that contains luminal (*n* = 689), basal-like (*n* = 169), normal-like (*n* = 35), and HER2+ (HER2-enriched, *n* = 75) BCs, respectively. The deceased cases were *n* = 132, which constituted 13.6% (132/969) of the subjects within the TCGA cohort. We thus combined the luminal A and luminal B cases into the luminal A + luminal B group and the basal-like and normal-like cases into the basal-like + normal-like group (*n* = 204). While both the DEG36 and SigDEG23 stratified the mortality risk of luminal BCs, the DEG36 was more effective (comparing Figure 8A to Appendix A). In the basal-like + normal-like population, the DEG36 and SigDEG23 displayed a robust stratification of the mortality risk (Figure 8B). The discrimination was at an AUC value of 0.83 (Figure 8C) and showed higher efficacy in patients who were deceased earlier at AUC values > 0.95 (Figure 8D). In both the luminal and basal-like + normal-like populations, the DEG36 and SigDEG23 predicted mortality after adjusting for age at diagnosis and tumor stage (Figure 8E). We further analyzed the DEG36′s biomarker potential by estimating cutoff points using the empirical, kernel, and normal methods with 1000 bootstraps with the R *cutpointr* package. The median ROC–AUC in both the in-bag and out-of-bag 1000 bootstrapping samples was 0.83 for all three methods (empirical, kernel, and normal). Importantly, effective stratification of BC fatality was achieved with a range of cutoff points (Appendix A); with the proper cutoff point estimated, a balanced sensitivity (70%) and specificity (70.1%) can be achieved (Appendix A). Collectively, the impressive performance in terms of the cutoff point estimation in both the in-bag and out-of-bag samples and the effective risk stratification achieved in a range of cutoff points (Appendix A) support the DEG36 as an effective biomarker in predicting the mortality risk of TNBCs or basal-like BCs.

The DEG36 performs better in basal-like BCs than claudin-low and normal-like BCs within the METABRIC dataset (Figure 6A). We thus analyzed the DEG36’s biomarker value in the basal-like subtype BCs (*n* = 169, event/deceased cases *n* = 22) within the TCGA cohort. With the cutoff point estimated using Maxstat, the DEG36 correctly separated 20 out of 22 deceased cases into the high-risk group (Figure 8F); the HR of the high-risk group was 21.8 (Figure 8F), i.e., patients in the high-risk group had a 21.8-fold higher risk of death compared to those in the low-risk group. Effective mortality risk stratification was also achieved with a range of cutoff points with high sensitivity and specificity (except for the cutoff point estimated using the normal method) (Figure 8G). The DEG36 discriminated the mortality of patients with basal-like BCs at an AUC of 0.93 at 13.3 months (Appendix A). Collectively, the above evidence supports the DEG36 as a novel and robust multigene panel in predicting the mortality risk of basal-like BCs, which is strengthened by the fact that the DEG36 was derived from a functional study without undertaking any modeling manipulations.

### 3.7. DEG36 Predicts the Exclusion of Cytotoxic Lymphocytes from BC

Given that evading the immune response is a hallmark of cancer, we anticipated a relationship between the DEG36 and immunotolerance in the tumor microenvironment (TME). To examine this possibility, we profiled the immune cell content associated with BC using the TCGA PanCancer BC RNA-seq dataset (*n* = 1082) with multiple computational programs, including xCell and ssGSEA [40], Epic [37], MCPCounter [38], quanTIseq [39], and CIBERSORT [36]. In the luminal BCs, significant reductions in CD8+ T cells and NK cells were detected in the high-risk group tumors stratified by the DEG36 compared to the tumors in the low-risk group (Figure 9A). Reductions in B cells and Tfh cells that facilitate B cell differentiation also occurred in the high-risk luminal BCs (Figure 9A). Concurrent increases in macrophages, M2-polarized macrophages, and Th2 cells were observed in the high-risk luminal BCs stratified by the DEG36 (Figure 9A). Infiltration of macrophages, M2 macrophages, and Th2 CD4+ T cells contributes to cancer immune escape [76,77]. Reductions in cytotoxic cells, B cells, and Tfh cells were observed in the high-risk TNBCs (basal-like + normal-like) stratified by the DEG36 (Figure 9B). Decreases in cytotoxic cells in the high-risk luminal BCs and reductions in CD8+ T cells and NK cells in the high-risk TNBCs were also demonstrated (Appendix A). Additionally, the high-risk TNBCs were associated with an elevation in mesenchymal stem cells (MSCs) (Figure 9B). MSCs enhance immune escape in cancers [78]. Collectively, the above evidence supports the exclusion of cytotoxic immune cells from the high-risk luminal BCs and TNBCs stratified by the DEG36.

The DEG36 is composed of human genes homologous to the mouse DEGs derived from 4T-1 tumors treated with taxifolin vs. DMSO, suggesting an impact of taxifolin treatment on the exclusion of cytotoxic immune cells. This concept is supported by a significant enrichment of CD8+ T cells in 4T-1 tumors treated with taxifolin in comparison to those treated with DMSO (Figure 9C,D). Taken together, these observations reveal the exclusion of CD8+ T cells as a potentially important contributor to the taxifolin-mediated attenuation of 4T-1 tumor growth.

## 4. Discussion

BC is the most commonly diagnosed cancer and the leading cause of cancer deaths in women, even with the available targeted therapies for ER+ and HER2+ BCs and the recent development of targeting TNBCs for their genome instability. Further advances in our understanding of BC and our ability to manage the disease remain an overarching task. BC pathogenesis and progression are affected by a dynamic and unbalanced ecosystem, including ROS-induced genomic damage, inflammation, and metabolic alterations favoring BC progression. In this regard, taxifolin as a well-established antioxidant nutritional supplement that possesses anticancer activities, including in BC. Nonetheless, the underlying mechanisms of taxifolin-derived anti-BC actions remain incompletely understood; as a result, its clinical potential has yet to be fully explored.

We provide several discoveries supporting taxifolin’s clinical applications in BC therapy. While taxifolin induced minimal alterations in the gene expression in 4T-1 allografts, these alterations are important, as they function in skin development and developmental-related events such as water homeostasis and cornified envelope (Figure 1H). Of note, HRNR, FLG2, CRCT1, and KPRP promote epidermal keratinocytes to undergo cornification, a form of terminal differentiation and programmed cell death required for the formation of the outmost skin barrier [44]. Intriguingly, HRNR is expressed at the cornified envelope [69]. Additionally, KLK7 (another taxifolin-induced DEG) also regulates cornification [79]. The *HRNR*, *FLG2*, *CRCT1*, and *KPRP* genes reside in 1q21.3, a chromosome region that was amplified in recurrent BC [29], and some genes within this amplicon promote BC progression [29]. For instance, S100A8 expression was increased in BCs with 1q21.3 amplification and contributed to BC [29]. However, the expression of HRNR, FLG2, CRCT1, and KPRP was not upregulated in BCs with a 1q21.3 copy number increase, suggesting the presence of unidentified mechanisms suppressing their expression to facilitate BC. While deciphering these mechanisms requires thorough investigations in future, epigenetic regulations, including promoter methylation, are likely among the potential mechanisms underpinning the suppression of the *HRNR*, *FLG2*, *CRCT1*, and *KPRP* genes in BC with 1q21.3 copy number increases.

Despite the mechanisms underlying the repression of *HRNR*, *FLG2*, *CRCT1*, and *KPRP* expression in BCs being unclear, upregulations of these genes by taxifolin, along with taxifolin’s action in reducing 4T-1 tumor growth, imply an appealing possibility to target recurrent BCs with 1q21.3 amplification by taxifolin. Given that these BCs harbor 1q21.3 amplification and the increased copy numbers of *HRNR*, *FLG2*, *CRCT1* and *KPRP* genes, these recurrent BCs are likely vulnerable to these genes-derived tumor suppressions. It is thus tempting to suggest that primary and recurrent BCs with 1q21.3 amplification can be treated with taxifolin or other mechanisms leading to their altered expression. The amplification of 1q21.3 occurs in all the BC subtypes [29], which is consistent with the amplification of the *HRNR*, *FLG2*, *CRCT1*, and *KPRP* genes in ER+, HER2+, and TN BCs (Figure 2A), suggesting that taxifolin or other mechanisms inducing HRNR, FLG2, CRCT1, and KPRP expressions could also be used as targeted therapies for TNBCs. Considering the general lack of targeted treatments for TNBCs, taxifolin and other mechanisms might be particularly appealing. This therapeutic potential is supported by the low levels of HRNR, FLG2, and KPRP protein expression in BC (Figure 2D and Appendix A), indicating that BC can express these proteins. Another therapeutic application lies in taxifolin’s ability to induce CD8+ T cell infiltration (Figure 9C,D), suggesting a combinational therapy involving immune checkpoint blockades (ICB) and taxifolin.

While the exact mechanisms responsible for taxifolin upregulating the *HRNR*, *FLG2*, *CRCT1* and *KPRP* genes and other component genes of the DEG36 remain to be explored in the future, it is tempting to suggest taxifolin’s action in regulating cell metabolism as being relevant. This suggestion is based on the well-studied antioxidant properties of taxifolin. It will be interesting to investigate the relationship between the upregulation of the above DEGs and the ROS status in tumor cells upon taxifolin treatment. Furthermore, the cell metabolic rate (fitness) can be reflected by the oxygen consumption rate (OCR) [80]. The impact of taxifolin on cancer cell metabolism can be measured by the Seahorse assay using its in vitro and ex vivo settings [81,82,83]. The potential contribution of taxifolin to *HRNR*, *FLG2*, *CRCT1*, *KPRP* and other tumor suppressor gene expressions in BC via its impact on cell metabolism is indeed intriguing, considering that metabolic alterations are a typical feature of cancer cells [84,85].

Our research also yielded a novel and robust multigene panel that predicted the poor prognosis of basal-like BCs. This panel was produced from 4T-1 tumors, a murine TN-like BC cell line. This panel thus does not share a direct linkage with either the METABRIC or TCGA cohort; nonetheless, the DEG36 robustly stratified the BC fatality, particularly in basal-like BCs, a major constituent of TNBCs [75]. In the TCGA cohort, approximately 1.2% of patients received neoadjuvant therapy prior to surgery, while pharmaceutical adjuvant therapy was given to 10.5% of patients after surgical resection (see Section 2.1). The DEG36 gene signature shows promise in assessing prognosis at the time of diagnosis or during the early stages of clinical management, potentially offering a broad application. In comparison, all the ER+ tumors were treated with endocrine therapy and almost 60% of patients received radiotherapy in the METABRIC cohort (see Section 2.1). In this regard, our multigene panel (DEG36) can be used to estimate treatment outcomes. This panel might have a unique clinical application in evaluating poor prognosis in TNBCs, a BC subtype with limited prognostic biomarkers.

## 5. Conclusions

We report here a unique potential of taxifolin in BC therapy. Taxifolin induced the upregulation of 36 genes in murine BC. Their human homologue genes formed a robust multigene panel (DEG36) that effectively predicted the mortality risk of basal-like BCs, a major molecular subtype of TNBCs. These DEGs consisted of multiple tumor suppressors and genes that negatively impacted BC. Among these genes, *HRNR*, *FLG2*, *CRCT1*, and *KPRP* promote cornification, a form of programmed cell death. In addition to possessing tumor-suppression functions, these taxifolin-induced genes (*HRNR*, *FLG2*, *CRCT1*, and *KPRP*) reside in 1q21.3 and are coamplified in recurrent BCs with 1q21.3 amplification. Our research thus revealed a critical vulnerability of BCs with 1q21.3 amplification, including in both primary and recurrent BCs. In the latter, evidence suggested the amplification to be common or up to 70%. Furthermore, we suggest that this vulnerability can be exploited by taxifolin and other potential mechanisms. In this research, we have consolidated the foundation for taxifolin’s application as a component in BC therapy and particularly in BCs with 1q23.1 amplification. Taxifolin may enhance the effectiveness of ICB therapy owing to the taxifolin-mediated enhancement of CD8+ T cell infiltration. We have constructed the DEG36 with a robust predictive power toward poor mortality in basal-like BC and discovered an important vulnerability in recurrent BCs. The knowledge presented here, which is related to 1q21.3-associated vulnerability and the biomarker potential of the DEG36, is innovative and should be further examined in both basic and clinical research settings. These research activities may significantly improve our ability to manage BC patients.

## Figures and Tables

**Figure 1 cancers-15-03203-f001:**
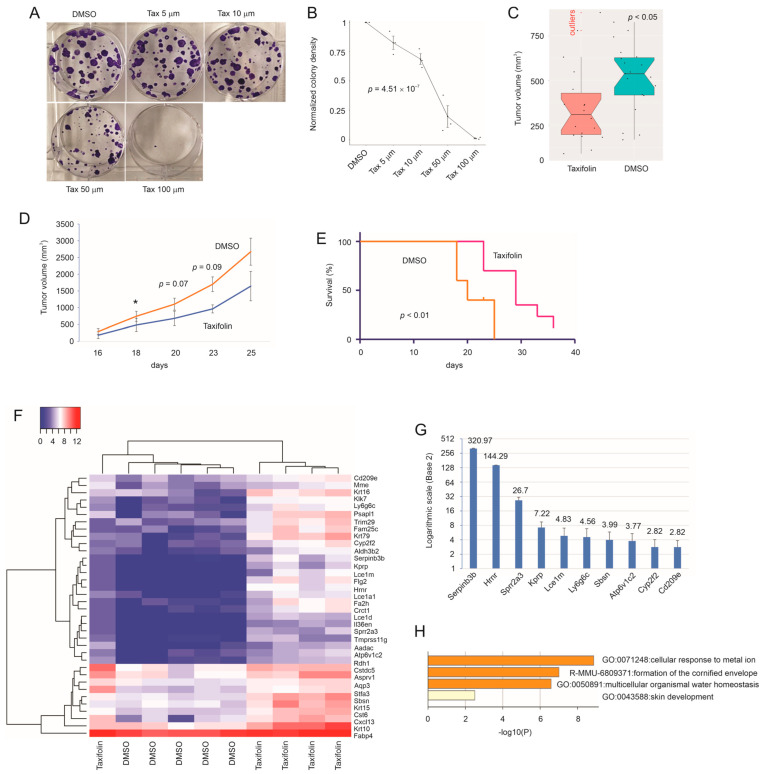
Taxifolin-mediated inhibition of 4T-1 breast tumor growth. (**A**,**B**) 4T-1 cells were seeded at 1000 per well and treated with DMSO or the indicated doses of taxifolin (Tax) for 10 days. Experiments were repeated 3 times. Typical images (**A**) and quantification (**B**) (mean ± SD/standard deviation) are shown. Statistical analysis was performed via one-way ANOVA. (**C***–***E**) BALB/cJ mice were s.c. implanted with 4T-1 cells (10^4^ cell/implantation) and treated with either DMSO or taxifolin (*n* = 20 per group) following an alternative 3- and 4-day schedule with the first treatment at 3 days post tumor implantation. Tumor volumes at day 20 (**C**) and throughout the experimental duration (**D**) are presented. * *p* < 0.05 compared to the taxifolin treatment by 2-tailed Student’s *t*-test. The Kaplan–Meier survival curve for reaching the endpoint (decline in health or tumor size) was constructed (**E**). Statistical analyses were performed via the Mann–Whiney U test (**C**), Student’s *t*-test (**D**), and log-rank test (**E**). (**F**) RNA-seq analysis of the 4T-1 tumors treated with DMSO (*n* = 5) or taxifolin (*n* = 5) was performed. The heatmap presents the expression of *n* = 36 DEGs. (**G**) Real-time PCR analysis of the indicated DEGs in all the tumors treated with DMSO and taxifolin. Gene expression in the taxifolin-treated tumors was presented as a fold change (mean ± SD) to the respective gene expression in the DMSO-treated tumors. *p* < 0.05 for all the comparisons via a 2-tailed Student’s *t* test. Fold changes for individual genes are presented on the top of the bars. (**H**) Pathway enriched among the *n* = 36 DEGs was analyzed using an overrepresentation analysis (ORA)-based enrichment analysis with the Metascape platform [35].

**Figure 2 cancers-15-03203-f002:**
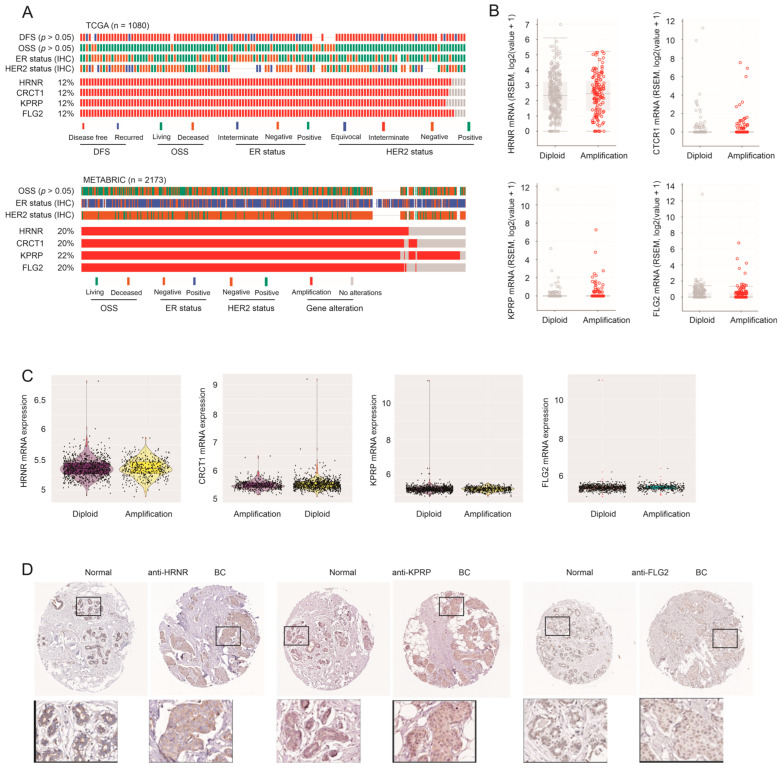
Amplification of genes located in 1q21.3 in breast cancer. (**A**) Amplification of the *HRNR*, *CRCT1*, *KPRP*, and *FLG2* genes residing in 1q23.1 in BCs within the TCGA and METABRIC datasets. The number of cases utilized in the analyses is indicated; only the proportion of tumors with the amplifications are shown. Images were constructed using tools provided by cBioPortal. DFS: disease-free survival; OSS: overall survival status. The indicated *p* values were produced using the Chi-square test for the associations between gene amplifications and DFS or OSS. (**B**,**C**) Boxplot of the indicated gene expressions in BCs without (diploid) and with their gene amplifications in the TCGA cohort (**B**) and METABRIC dataset (**C**). Graphs in panels (**B**,**C**) were produced using tools provided by cBioPortal and R, respectively. Statistical analyses were performed using the Mann–Whiney U test. No meaningful differences were detected in the indicated comparisons. (**D**) IHC images of normal breast and BC tissues stained for HRNR, KPRP, and FLG2 were downloaded from the Human Protein Atlas (https://www.proteinatlas.org/; accessed on: 11 November 2022). The marked regions were enlarged 3.5 fold.

**Figure 3 cancers-15-03203-f003:**
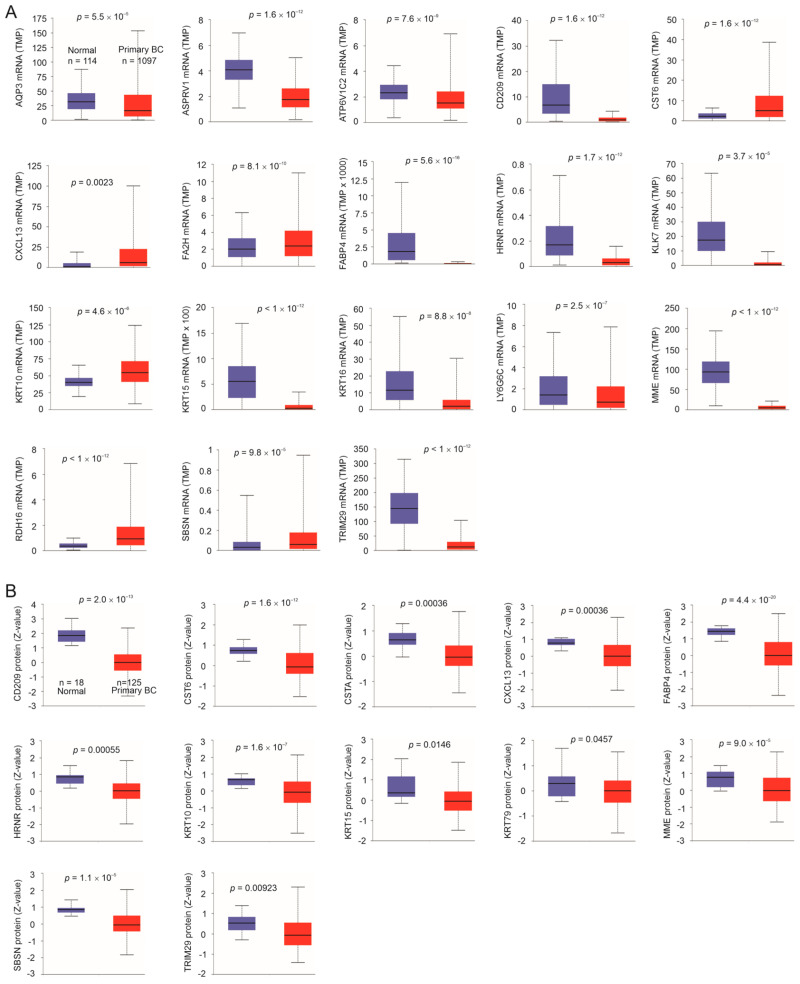
Downregulation of taxifolin-induced DEGs. Analysis of the human counterparts of the taxifolin-induced murine genes in 4T-1 tumors. Analyses were performed using the TCGA dataset organized by UALCAN [34]. (**A**) mRNA expression of the indicated genes. (**B**) Protein expression of the indicated genes.

**Figure 4 cancers-15-03203-f004:**
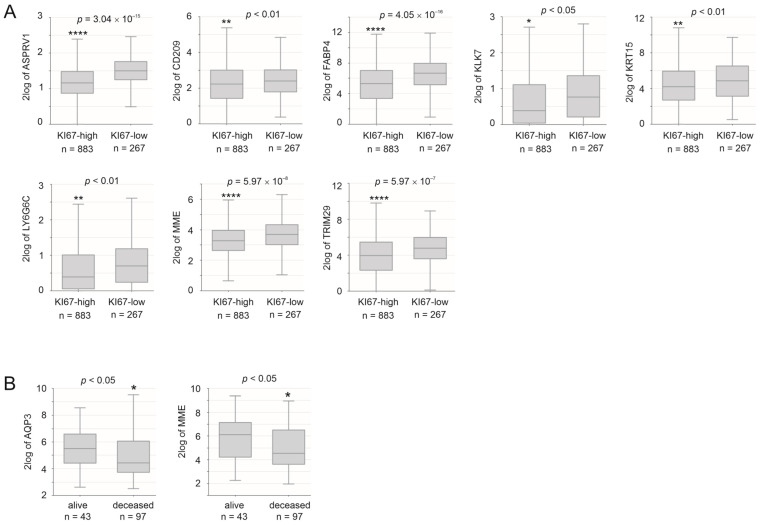
Association of taxifolin-affected DEGs with BC proliferation and prognosis. Analysis was performed using the R2: Genomics Analysis and Visualization Platform. (**A**) The mRNA expressions of the indicated genes in BCs in the Ki67-high and -low groups within the Gruvberger-Saal dataset; patients in this cohort were treated with endocrine therapy (78%) and chemotherapy (39.2%) [72]. (**B**) AQP3 and MME mRNA expressions in the indicated BC groups within the Sinn dataset; patients in this cohort were treated with endocrine therapy (69%) and chemotherapy (24%) [73]. *: *p* < 0.05; **: *p* < 0.01, and ****: *p* < 0.0001.

**Figure 5 cancers-15-03203-f005:**
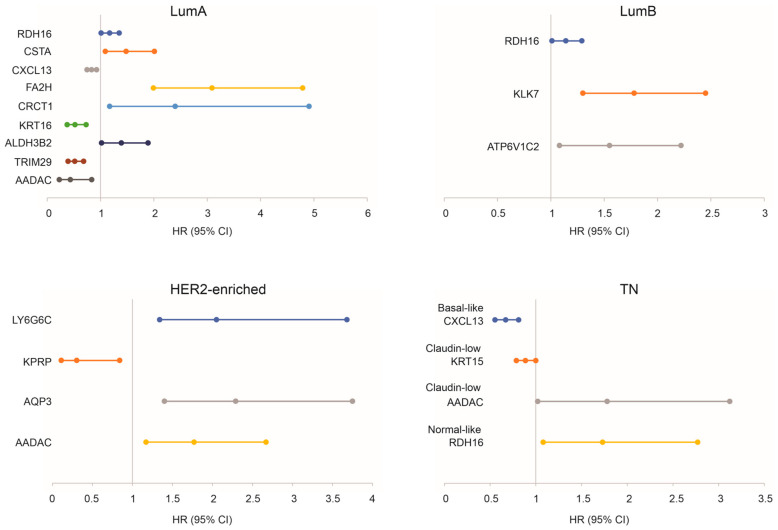
Prediction of mortality by taxifolin-induced genes. Prediction of BC mortality by the indicated genes in the indicated intrinsic BC subtypes within the METABRIC dataset. Analyses related to the basal-like, claudin-low, and normal-like BCs are presented in the TN panel. LumA and LumB: luminal A and luminal B.

**Figure 6 cancers-15-03203-f006:**
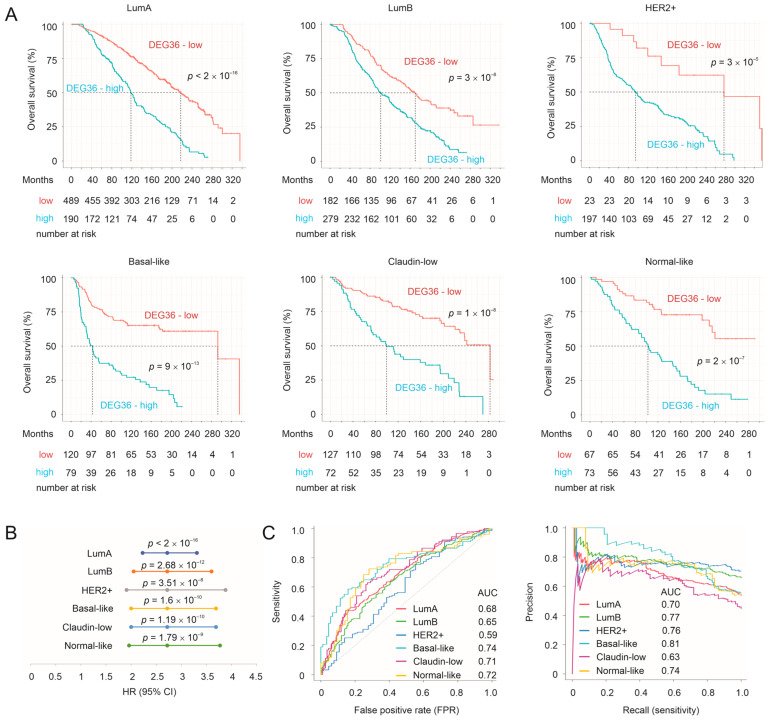
Stratification of BC mortality risk. (**A**) The DEG36 risk scores were calculated for individual tumors. Cutoff points were estimated using Maximally Selected Rank Statistics with the R Maxstat package. The Kaplan–Meier curves and log-rank test were carried out with the R survival package. (**B**) Prediction of OS in the indicated molecular subtypes of BCs (HER2+: HER2-enriched). (**C**) ROC–AUC (receiver operating characteristic–area under the curve) and precision recall (PR)–AUC curves for the discrimination of BC mortality in the METABRIC cohort.

**Figure 7 cancers-15-03203-f007:**
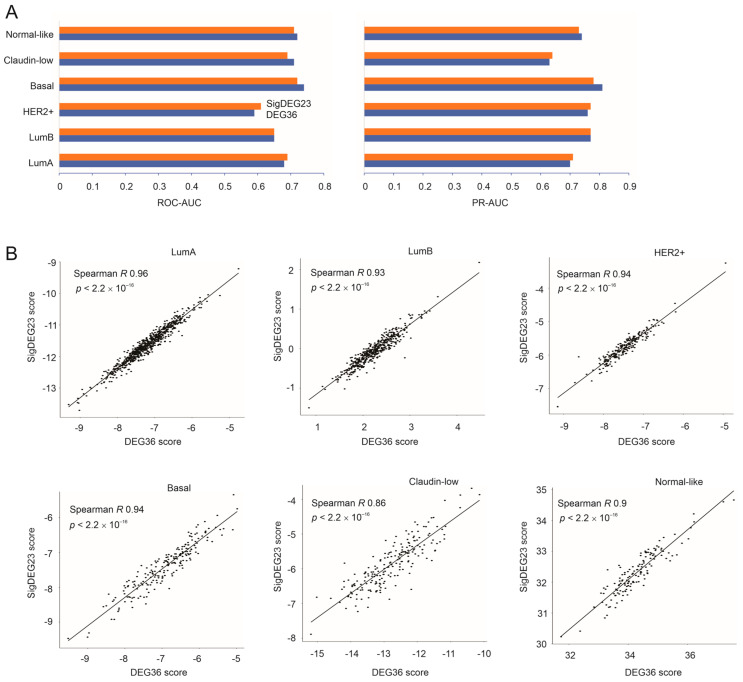
Equivalent prognostic biomarker potential between the DEG36 and SigDEG23. (**A**) ROC-AUC and PR-AUC curves of the DEG36 and SigDEG23 for the discrimination of BC mortality in the indicated intrinsic subtypes (Basal: basal-like; HER2+: HER2-encriched) within the METABRIC. Color indicators for the SigDEG23 and DEG36 bars are shown (see the HER2+ bars). (**B**) Correlations between the SigDEG23 and DEG36 risk scores in the indicated intrinsic subtypes of BC within the METABRIC.

**Figure 8 cancers-15-03203-f008:**
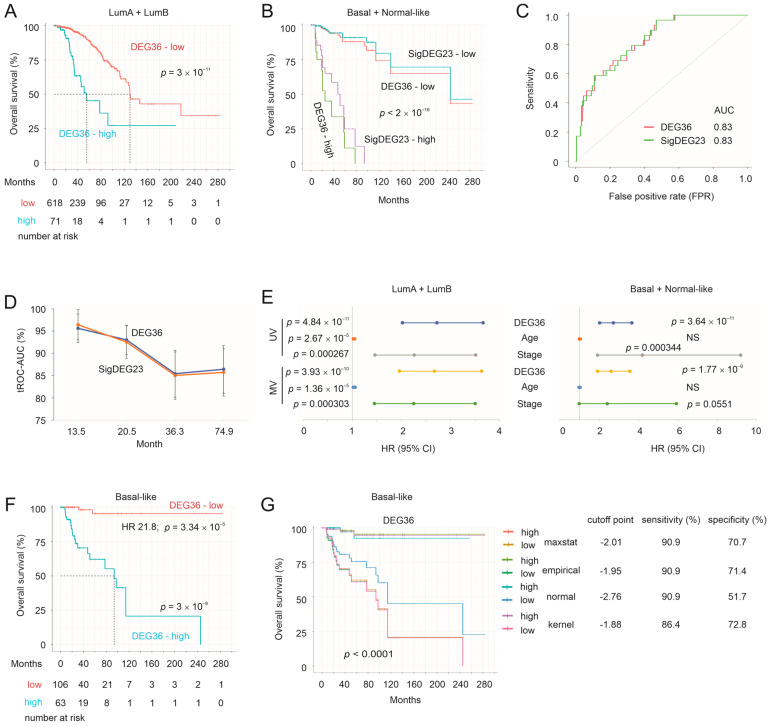
Validation of the DEG36 and SigDEG23 in the TCGA dataset. The TCGA PanCancer Atlas BC dataset was used for the validation analyses. (**A**,**B**) Luminal A (LumA) and luminal B (LumB) BCs were combined into the LumA + LumB group; basal-like (Basal) and normal-like BCs were combined into the basal + normal-like group. Stratification of poor prognosis in the indicated BC groups with the DEG36 and SigDEG23. (**C**) ROC-AUC curves. (**D**) Time-dependent ROC-AUC curves for the DEG36 and SigDEG23 for the discrimination of poor OS. (**E**) Univariate (UV) and multivariate (MV) Cox analysis-derived HR and 95% CI in the prediction of survival probability. NS: not significant. (**F**) Stratification of the BC fatality risk by the DEG36 in basal-like BCs within the TCGA dataset. The HR for patients in the high-risk group compared to those in the low-risk group is indicated. (**G**) Cutoff points of the DEG36 risk score in separating high- and low-mortality risk BCs were estimated using the Maxstat, empirical, normal, and kernel methods. Individual survival curves produced via individual cutoff points and the associated specificity and sensitivity are presented. Statistical tests were performed using the log-rank test.

**Figure 9 cancers-15-03203-f009:**
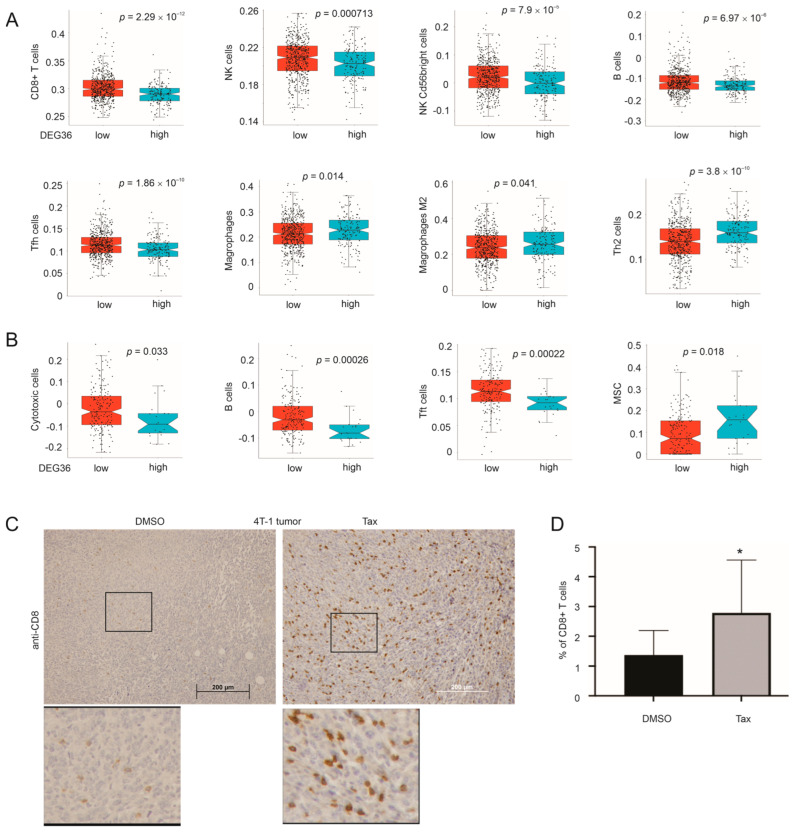
The effects of the DEG36 and taxifolin treatment on immune cell infiltration in BC. (**A**,**B**) Immune cell contents in the LumA + LumB (**A**) and basal-like + normal-like (**B**) populations within the TCGA cohort. The BC-associated immune cell populations were determined using ssGSEA [40]. (**C**,**D**) 4T-1 tumors treated with DMSO (*n* = 10) or taxifolin (*n* = 10) were IHC stained for CD8 using the anti-CD8 antibody; average *n* = 6075 cells per DMSO-treated tumor and *n* = 12,872 cells per taxifolin-treated tumor were counted. The control IgG did not produce a detectable signal in either the DMSO- and taxifolin-treated tumors (Appendix A). Typical images with 3-fold enlargement for the indicated regions and quantification (mean + SD) are included. * *p* < 0.05 in comparison to DMSO treatment via a 2-tailed Student’s *t*-test. Tax: taxifolin.

**Table 1 cancers-15-03203-t001:** Taxifolin-upregulated DEGs and their reported roles in BC.

Symbol ^i^	Description	Locus	Pub ^ii^	Function in BC ^iii^	Ref
FAM25A ^vi^	Family with sequence similarity 25 member A	10q23.2	0	Unknown	
CST6	Cystatin E/M	11q13.1	37	Tumor suppressor activities	[45]
ALDH3B2 ^vi^	Aldehyde dehydrogenase 3 family member B2	11q13.2	3	A component gene in a multigene biomarker of breast cancer	[56]
TRIM29 ^vi^	Tripartite motif containing 29	11q23.3	13	A tumor suppressor of BC	[46]
KRT79 ^vi^	Keratin 79	12q13.13	0	Unknown	
RDH16 ^vi^	Retinol dehydrogenase 16	12q13.3	1	A component gene in a multigene biomarker of Her2+ breast cancer	[57]
FA2H ^vi^	Fatty acid 2-hydroxylase	16q23.1	9	Downregulation in TN BC; tumor-suppressive actions in BC	[58]
KRT10	Keratin 10	17q21.2	6	expressed in breast cancer cells	[59]
KRT15	Keratin 15	17q21.2	12	Association with good prognosis in BC	[48]
KRT16 ^vi^	Keratin 16	17q21.2	7	Association with poor prognosis in BC	[60]
SERPINB3	Serpin family B member 3	18q21.33	3	Facilitating BC	[61]
CD209 ^iv^	CD209 molecule	19p13.2	8	In a multigene biomarker of BC	[62]
SBSN ^vi^	Suprabasin	19q13.12	0	Unknown	
CYP2F1 ^vi^	Cytochrome P450 family 2 subfamily F member 1	19q13.2	2	Its expression is potentially affected by estrogen in MCF-10F cells	[63]
KLK7 ^vi^	Kallikrein-related peptidase 7	19q13.41	19	Favorable prognosis in TNBC	[49]
HRNR ^vi^	Hornerin	1q21.3	1	Reductions in high T stage, LN metastasis, and BC cells with metastatic capacity	[64]
CRCT1	Cysteine rich C-terminal 1	1q21.3	0	Unknown	
KPRP ^vi^	Keratinocyte proline-rich protein	1q21.3	1	Mutations of KPRP in TNBCs with pathological response to neoadjuvant trials	[65]
FLG2 ^vi^	Filaggrin 2	1q21.3	2	Mutations in 4/12 inflammatory BC	[66]
ASPRV1 ^vi^	Aspartic peptidase retroviral like 1	2p13.3	0	Unknown	
ATP6V1C2 ^vi^	ATPase H+ transporting V1 subunit C2	2p25.1	1	Facilitating BC	[67]
IL36RN ^vi^	Interleukin 36 receptor antagonist	2q14.1	0	Unknown	
CSTA ^vi^	Cystatin A	3q21.1	14	An ER target gene and a suppressor of BC	[47]
AADAC ^vi^	Arylacetamide deacetylase	3q25.1	0	Unknown	
MME ^vi^	Membrane metalloendopeptidase	3q25.2	27	High expression in dormant BC cell residing in the bone and lung; BCs with high MME expression display better OS.	[50,51]
PSAPL1 ^iv^	Prosaposin-like 1	4p16.1	1	In a multigene biomarker of BC	[68]
TMPRSS11E ^v,vi^	Transmembrane serine protease 11E	4q13.2	0	Unknown	
CXCL13 ^vi^	C-X-C motif chemokine ligand 13	4q21.1	75	Facilitating immune response; association with favorable OS	[52,53]
LY6G6C ^vi^	Lymphocyte antigen 6 family member G6C	6p21.33	0	Unknown	
FABP4	Fatty acid binding protein 4	8q21.13	55	Facilitating BC progression	[54]
AQP3 ^vi^	Aquaporin 3 (Gill blood group)	9p13.3	30	Association with poor OS in TNBC	[55]

i: human homologous genes; ii: number of publications relevant to BC listed in PubMed on 26 October 2022; iii: published roles; iv: not present in the METABRIC cohort; v: not present in the TCGA PanCancer Atlas BC cohort; and vi: component genes of SigDEG23, a panel optimized from the DEG36.

## Data Availability

All materials are available upon request.

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
