# Peer review of "Taxifolin Inhibits Breast Cancer Growth by Facilitating CD8+ T Cell Infiltration and Inducing a Novel Set of Genes including Potential Tumor Suppressor Genes in 1q21.3"

_cancers, 2023, doi:10.3390/cancers15123203_

Round 1

Reviewer 1 Report

Review of:

  Taxifolin inhibits breast cancer growth by facilitating CD8+ T cell infiltration and inducing a novel set of genes including potential tumor suppresser genes in 1q21.3

 Comments to the Authors

The authors claim that Taxifolin induces expression of a gene signature responsible to suppress breast cancer.

In their first set of experiments, tumor-bearing mice are treated with Taxiflin and observe reduction of tumor growth. The authors then identified 36 upregulated genes in Taxifolin treated tumors and matched them tu human orthologues,  focusing on 4 genes and a set of 23 human genes, to stratify distinct subtypes of BC by the overall survival, hazard ratio. The authors claimed to have found prognostic biomarkers for BC using Taxifolin.

I think this data would be of great interest for the research community but unfortunately the data has not been shared on a public data sharing platform. This needs to be addressed foremost.

 Detailed comments:

Line 197, (Fig1A,B) several Taxifolin dosages are tested, A IC50 calculation would be great.

On line 197 they mention Matrigel but there is no data about it, I would remove that.

On line 201 104, 105, and 5*105 cells per implantation are mentioned. Again, there is no data comparing this. I would remove the amount not used.

Line 203, Fig S1 please include day -4. Looks like Taxifolin mice gained more wight in those 8 days. (days -4 to day 40 Please clarify.

Line 206, Fig 1C. A better graph would be a XY plot with tumor progress over time +- Taxifolin.

Line 206, Fig. 1D. What is considered survival, were the mice euthanized once the tumor reached a certain size?

Line 210, Please deposit the RNAseq data to a public database, or add the full differential expression analysis result as a supplement, so the reader can decide why those 36 genes have been chosen. FABP4 looks like there is not much change.

Line 213 Fig 1F, why only these genes were tested on the qPCR and not all the four favored genes (HRNR, CRCT1, KPRP, and FLG2)? Additionally, please add also the qPCR results of the RNA expression levels relative to housekeeping genes such as bActin or GAPDH.

Lines 213 – 219, What enrichment analysis was done here? Broad institute’s GSEA or others would have resulted in many more genes sets and/or pathways. The fact that these are exclusively skin GO shows me that the analysis of the 4T1 tumors which were SQ injected contained skin tissue. Can you rule that out? Again, the full DEG analysis is needed.

Table 1, make it more obvious which genes are in the SigDEG23 group

Line 300, please show this data in the supplement

Line 315, Fig 2D, S3C please add a quantification/statistics of the microarray, what is the classification of the BC? Is there any treatment in these cohorts? Need more information about this.

Figure 3, 4,  what is the classification of these BCs? Is there any treatment in these cohorts? Need more information about this.

Figure 5, 6, 7, 8, 9,  Is there any treatment in these cohorts? Need more information about this.

Line 459, please show this data in the supplement

minor typos along manuscript, need more proof reading

here are some:

line 287 the rest of THE three genes

line 396, Fig 6 legend: Mortality instead of Morality

Line 541 METABRIC instead of MEBRIC

Author Response

We appreciate the reviewer’s overall positive comments and insightful remarks. Here are our detailed revisions.

“Line 197, (Fig1A,B) several Taxifolin dosages are tested, A IC50 calculation would be great.”

Authors' response – The IC50 was calculated and presented (line 199, marked with red).

“On line 197 they mention Matrigel but there is no data about it, I would remove that.”

Authors' response – We agree and have removed the Matrigel statement (see line 199 marked with red, which contains no “Matrigel”).

“On line 201 104, 105, and 5*105 cells per implantation are mentioned. Again, there is no data comparing this. I would remove the amount not used.”

Authors' response – The statements involving different number of cells have been deleted and the focus is on the dose of 104 cells used in this research (lines 202-204, marked with red).

“Line 203, Fig S1 please include day -4. Looks like Taxifolin mice gained more wight in those 8 days. (days -4 to day 40 Please clarify.”

Authors' response – Thank the reviewer for pointing out the lack of details in Figure S1. In our study, the mice were randomly divided into DMSO and taxifolin (Tax) group at day 0, followed by first treatment at day 4. Therefore, weight changes observed between day 0 and day 4 in both groups were base-line alterations unrelated to the treatment. We performed statistical analyses and found no significant differences in weight gains between DMSO and Tax treatment groups throughout the entire experimental duration. We have revised Figure S1, included additional experimental details in the legend, and reworked the manuscript (lines 204-205, marked with red) to address the concern. We trust that these modifications adequately addressed the issues raised by the reviewer.

“Line 206, Fig 1C. A better graph would be a XY plot with tumor progress over time +- Taxifolin.”

Authors' response – We appreciate this insightful remark. A new panel (Figure 1D) was arranged to show the profiles of tumor growth in the DMSO- and Tax-treated groups. We noticed a wide range of variations in the growth of 4T-1 syngeneic tumors, which was in part due to distant metastasis. It is important to note that in this animal model, primary 4T-1 tumors often exhibit slow growth along with concurrent distant metastasis. In this regard, the growth profile of primary 4T-1 tumors may not accurately reflect the impact of taxifolin on tumorigenesis. This is a concern as 4T-1 tumors in the DMSO group were more likely to metastasize to other sites compared to those in the Tax group. Nonetheless, our research was not specifically optimized to address the metastasis issue, as it would require a substantially larger number of animals. To overcome this limitation in the current experimental setting, we performed median-based analysis of tumor volume at a particular time point (Figure 1C) and survival analysis (Figure 1D in the last submission and Figure 1E in this revision). These additional analyses provide a more comprehensive understanding of the impact of taxifolin on 4T-1 tumorigenesis. We have included these key messages in this revision (lines 208-218, marked with red). The inclusion of the tumor growth profile (Figure 1D this revision) in conjunction with Figure 1C and Figure 1E (survival curve) offer a more precise depiction of the effect of Tax on 4T-1 tumorigenesis, for which we thank Reviewer #1 for the comments.

“Line 206, Fig. 1D. What is considered survival, were the mice euthanized once the tumor reached a certain size?”

Authors' response – The corresponding panel is Fig 1E in this revision. The endpoint was defined as a decline in health and/or tumor size; the details of endpoint setting have been added in this revision (lines 217-218, marked with red and line 236 marked with red).

“Line 210, Please deposit the RNAseq data to a public database, or add the full differential expression analysis result as a supplement, so the reader can decide why those 36 genes have been chosen. FABP4 looks like there is not much change.”

Authors' response – We agree! Data sharing is an essential aspect of research, and we are committed to promoting transparency and collaboration in our scientific community. We will deposit our RNA seq data shortly to a public database and make the data available to facilitate further analysis and discussion of our research findings. Hope this arrangement will address Reviewer #1’s concern.

“Line 213 Fig 1F, why only these genes were tested on the qPCR and not all the four favored genes (HRNR, CRCT1, KPRP, and FLG2)? Additionally, please add also the qPCR results of the RNA expression levels relative to housekeeping genes such as bActin or GAPDH.”

Authors' response – The corresponding panel is Fig 1G in this revision. The panel contains upregulation of mouse Hrnr and Kprp genes. For both murine Crct1 and Flg2 genes, we had challenges to amplify both by PCR. These difficulties could be attributed to various factors, including the need for primer optimizations. While we did not extensively pursue solving this issue, we focused on confirming the upregulation of Hrnr and Kprp genes in tumors treated with Tax. As requested by the Reviewer, we have organized the PCR amplification profiles for individual genes and actin in Table S2. We hope this clarification adequately addresses the reviewer’s concern.

“Lines 213 – 219, What enrichment analysis was done here? Broad institute’s GSEA or others would have resulted in many more genes sets and/or pathways. The fact that these are exclusively skin GO shows me that the analysis of the 4T1 tumors which were SQ injected contained skin tissue. Can you rule that out? Again, the full DEG analysis is needed.”

Authors' response – The type of enrichment was overrepresentation-based analysis performed on the n = 36 DEGs. The details have been added (line 241-243, marked with red). Regarding the GO enrichment relevant to skin process, we share the Reviewer’s concern. Nonetheless, it is important to note that subcutaneous (s.c.) tumor implantation, which is the most commonly used tumor model, is not known to be associated with skin processes. In our own research using the s.c. model on various types of cancer cells (prostate cancer, renal cancer, breast cancer, and lung cancer cells) combined with RNA seq, we have not observed a consistent involvement of skin processes. While we acknowledge that this issue should be further investigated in future studies, we believe that it is unlikely to be a significant factor in the current context.

“Table 1, make it more obvious which genes are in the SigDEG23 group”

Authors' response – SigDEG23 genes are clearly indicated in the revised Table 1 (see the footnote “vi”, lines 259-260, marked with red).

“Line 300, please show this data in the supplement”

 Authors' response – The data is presented in Figure S3, a new figure (see the corresponding line 312, marked with red).

“Line 315, Fig 2D, S3C please add a quantification/statistics of the microarray, what is the classification of the BC? Is there any treatment in these cohorts? Need more information about this.”

Authors' response – Data presented in Fig 2D and S3C were derived from the Human Protein Atlas (https://www.proteinatlas.org/) website. The tumor status was provided by experts from the website. There are no classifications on breast cancer subtypes and the limited number of samples available does not support a meaningful quantification. In this regard, the message provided by both figure panels is to provide a proof-of-principle demonstration, in attempt to illustrate the potential expression patterns and localization of the proteins of interest in breast cancer.

“Figure 3, 4,  what is the classification of these BCs? Is there any treatment in these cohorts? Need more information about this.”

Authors' response – Figure 3 addresses gene expression at both mRNA and protein levels with respect to normal vs tumor tissues; the details of gene expression in different subtypes of BC are not analyzed here. This research setting was chosen to maintain a focused approach and address the primary objective of our study. Figure 3 was derived from the TCGA cohort; the treatment details have been included (lines 107-109, marked with red). Treatment details for the cohorts used in Figure 4 are also provided (lines 361-363, marked with red).

“Figure 5, 6, 7, 8, 9,  Is there any treatment in these cohorts? Need more information about this.”

Authors' response – These figures are based on data from the TCGA and METABRIC cohorts; the treatment details for both cohorts are added (lines 105-109, marked with red). Furthermore, we have addressed the relevance of this knowledge in our research (lines 577-584, marked with red), which highlights how the inclusion of these cohorts adds value to this research. We believe these revisions have strengthened the overall discussion and we thank Reviewer #1 for this comment.

“Line 459, please show this data in the supplement”

Authors' response – We trust “Line 459” should be “Line 495”. A new supplementary figure (Figure S10) has been organized. Also see line 511 (marked with red).

“minor typos along manuscript, need more proof reading”

Authors' response – We have made a thorough effort to eliminate typos.

“line 287 the rest of THE three genes”

Authors' response – Added it (line 299, marked with red).

“line 396, Fig 6 legend: Mortality instead of Morality”

Authors' response – Corrected it (line 411, marked with red).

“Line 541 METABRIC instead of MEBRIC”

Authors' response – Corrected it (line 575, marked with red).

Reviewer 2 Report

In this manuscript, authors used 4T1 cell line and allogenic murine model to study the anti-tumorigenic effect of a flavonoid compound Taxifolin. Using bulk RNA sequencing in taxifolin treated tumor as compared to control group, authors identified gene signature of 36 differentially expressed genes that are further explored as a biomarker set to determine human breast cancer patient survival. Mechanistically, this study also demonstrates that taxifolin treatment increased CD8+ T cell infiltration in the tumor microenvironment of primary breast tumor and thus reduce immune suppression.

Overall, the study is interesting, provides important insight and clinical relevance in using DEG36 as set of prognostic biomarkers for breast cancer patient stratification. However, whole approach of using Taxifolin as anti-tumor agent, needs extensive experimental validation and several points need revision and further clarification to justify further consideration.

Major comments:

1.    Please confirm the dose of Taxifolin utilized for in vivo studies. Rectify the discrepancy between 50 mg/g or mg/kg in result section (e.g., page 5 and in legend for Fig S1).

2.    Although murine breast cancer models generated by ectopic injections (sub-cutaneous) of tumor cells are acceptable in the field, generating the allogenic mouse model by orthotopic implantation of tumor cells in the mammary fat pad is more reliable and accurate approach and provide less variability. Please explain if there is any specific reason why author used s.c. injection instead of orthotopic implantation of 4T1 tumor cells in the mammary fat pad to generate allogenic murine model for in vivo studies.

3.    For Fig. 1C, please provide tumor growth curve of this experiment for treatment vs. control group. If possible, provide growth curve of individual tumor rather just quantifying tumor volume at the end point to clearly associate the effect of taxifolin treatment on tumor progression and its anti-proliferative function.

4.    It would be interesting to check the effect of taxifolin treatment in established tumor to compare its relevance with clinical setting.

5.    In fig. S1, please explain: are 4T1 injected tumor bearing mice used in experiment for weight loss study or it’s a separate group of non-tumor bearing mice only used for toxicity study? Please provide p value in the figure legend.

6.    Fig 1E, F, G data does not directly show tumor suppression so avoid using statements that have strong claim without experimental support p8-9 line 218-219, 258-259. Mere enrichment of pathways related with programmed cell death (Fig 1G) does not prove the tumor suppression upon taxifolin treatment. Please provide experimental proof such as cleaved caspase 3 expression by IHC or flow analysis between control vs. treated tumors.

7.    Please describe/discuss the possible reason, why HRNR, CRCT1, KPRP, and FLG2 proteins were not upregulated even upon 30-70% amplification of gene? Is there any aberrant epigenetic or posttranslational regulation of these proteins associated with the BC?

8.    Please discuss relevance of Fig 5 in detail.

9.    Please explain in the material and method section or in figure legend, how risk score and cutoff values were determined for fig 6?

10. Fig. 9C, please provide high resolution representative IHC images. The image provided for control group is blurry. In addition, please provide details, how the scoring of IHC was done. Fig 9D shows % of CD8+ T cells, however, in method section it is mentioned to be quantified as H-score. Besides, please provide the levels of exhausted T cell and regulatory T cell fractions in addition to CD8+ or CD4+ T cells, as their presence in the tumor is key to determine the immunosuppressive microenvironment.

11. I would recommend testing combination therapy of taxifolin together with the checkpoint inhibitor like anti-PDL1. As low immune infiltration is one of the major causes of low efficacy of immunotherapy in solid tumors like breast cancer. If taxifolin increases the immune infiltration in tumor as indicated, it would be interesting to see if taxifolin treatment can increase the sensitivity of breast tumors for immunotherapy in combination treatment.

Minor comments:

1.    Many grammatical and typo errors were found throughout manuscript. Please correct the spelling of ‘suppressor’ in the title of the manuscript.

2.    Please provide scale bar for Fig 2D and S3C IHC images or indicate the magnification in figure legend.

3.    Error bars need to be defined together with a measure of central tendency for all the box plots used in the manuscript figures.

4.    Figure reference for given result is not matching in page 11, line 327; ‘Additionally, CSTA and KRT79 were only 326 downregulated at the protein level in BC (Figure 4B)’. Should it be Fig 3B? Please confirm.

5.    Please mention how statistical analysis was performed for Fig 8 in figure legend.

The manuscript require little bit reframing of sentences for better presentation. However, overall quality is good and acceptable. Few places have grammatical and typo-errors that needs to be corrected.

Author Response

We appreciate the reviewer’s overall positive comments and insightful remarks. Here are our detailed revisions.

Major comments

  1. “Please confirm the dose of Taxifolin utilized for in vivo studies. Rectify the discrepancy between 50 mg/g or mg/kg in result section (e.g., page 5 and in legend for Fig S1).”

     Authors' response – We apologize for these errors. Both have been corrected to mg/kg (line 200, marked with red; also see Figure S1 legend).

  1. “Although murine breast cancer models generated by ectopic injections (sub-cutaneous) of tumor cells are acceptable in the field, generating the allogenic mouse model by orthotopic implantation of tumor cells in the mammary fat pad is more reliable and accurate approach and provide less variability. Please explain if there is any specific reason why author used s.c. injection instead of orthotopic implantation of 4T1 tumor cells in the mammary fat pad to generate allogenic murine model for in vivo studies.”

     Authors' response – We share the Reviewer’s points regarding subcutaneous (s.c.) and orthotopic tumor models. We acknowledge that the mammary fat pad orthotopic tumor models would have been an ideal choice if the conditions had allowed for its implementations. This study was conducted during the challenging period of the pandemic, during which access to technician, animal care staff, and necessary materials for the orthotopic tumor model were limited or unavailable. As a result, we had to opt for the s.c. tumor model, which was feasible under the circumstances. Nonetheless, as the reviewer pointed out, s.c. tumor model remains the most widely used animal model in cancer research and provides relevant insights into tumorigenesis. With that, we trust Reviewer #2 would agree that the s.c. model used here was relevant and valuable. However, we are planning to carry out additional research using the mammary fat pad BC model in the near future.

  1. “For Fig. 1C, please provide tumor growth curve of this experiment for treatment vs. control group. If possible, provide growth curve of individual tumor rather just quantifying tumor volume at the end point to clearly associate the effect of taxifolin treatment on tumor progression and its anti-proliferative function.”

     Authors' response – We thank the reviewer for these insightful remarks. Similar comments were also raised by Reviewer #1. Here are our responses.

     A new panel (Figure 1D) was arranged to show the profiles of tumor growth in the DMSO- and Tax-treated groups. We noticed a wide range of variations in the growth of 4T-1 syngeneic tumors, which was in part due to distant metastasis. It is important to note that in the 4T-1 model, primary tumors often exhibit slow growth accompanied by concurrent distant metastasis. In this regard, the growth profile of primary 4T-1 tumors may not reveal the impact of taxifolin on tumorigenesis. This is a concern as 4T-1 tumors in the DMSO group were more likely to metastasize to other sites compared to those in the Tax group. We acknowledge this limitation, as addressing the metastasis issue would require a large number of animals and a more optimized experimental design. To overcome this limitation in the current experimental setting and provide a more comprehensive assessment, we performed a median-based analysis of tumor volume at a particular time point (Figure 1C) and survival analysis (Figure 1D in the last submission and Figure 1E in this revision). These messages are included in this revision (lines 208-218, marked with red). These additional analyses, combined with the tumor growth profile (Figure 1D this revision) and survival curve (Figure 1C and Figure 1E), allow for a more precise evaluation of the impact of Tax on 4T-1 tumorigenesis, for which we thank both Reviewer #1 and #2 for the comments.

  1. “It would be interesting to check the effect of taxifolin treatment in established tumor to compare its relevance with clinical setting.”

     Authors' response – We agree! However, it might be more proper to systemically analyze this task in future.

  1. “In fig. S1, please explain: are 4T1 injected tumor bearing mice used in experiment for weight loss study or it’s a separate group of non-tumor bearing mice only used for toxicity study? Please provide p value in the figure legend.”

     Authors' response – We have revised Figure S1 to clearly outline the study that was performed on tumor-free mice or for toxicity study. P values are also included.  

  1. “Fig 1E, F, G data does not directly show tumor suppression so avoid using statements that have strong claim without experimental support p8-9 line 218-219, 258-259. Mere enrichment of pathways related with programmed cell death (Fig 1G) does not prove the tumor suppression upon taxifolin treatment. Please provide experimental proof such as cleaved caspase 3 expression by IHC or flow analysis between control vs. treated tumors.”

     Authors' response – The corresponding panels are Figure 1F, G, and H in this revision. We agree with the reviewer for the need for precise interpretation and therefore have toned down the relevant statements to ensure accuracy (lines 223, 229, and 270-271; marked with red).

     We suggested keratinocyte cornification, a form of terminal differentiation and programmed cell death, could potentially contribute to the inhibition of BC by taxifolin. However, we acknowledge that cornification is different from apoptosis, i.e. cornification (although this is a type of programmed cell death) is unlikely the apoptotic form of programmed cell death. In this regard, we hope that Reviewer #2 would agree examination of caspase 3 cleavage is not an urgent requirement for this study.

  1. “Please describe/discuss the possible reason, why HRNR, CRCT1, KPRP, and FLG2 proteins were not upregulated even upon 30-70% amplification of gene? Is there any aberrant epigenetic or posttranslational regulation of these proteins associated with the BC?”

     Authors' response – We share the Reviewer’s insights; the lack of increases of mRNA expression for these genes in BCs with increases in their gene copies is intriguing; the underlying mechanisms should be investigated in future. This research may have great potential in BC therapy. In this revision, we have provided some thoughts on this issue (lines 539-544, marked with red). We hope that these additional considerations contribute to the overall discussion and help stimulate further research in this area.

  1. “Please discuss relevance of Fig 5 in detail.”

     Authors' response – We have briefly extended the discussion (lines 369-371, marked with red).

  1. “Please explain in the material and method section or in figure legend, how risk score and cutoff values were determined for fig 6?”

     Authors' response – Details on risk score calculation (lines 128-130, marked with red) and cutoff point estimation (lines 411-412, marked red) have been included.

  1. “Fig. 9C, please provide high resolution representative IHC images. The image provided for control group is blurry. In addition, please provide details, how the scoring of IHC was done. Fig 9D shows % of CD8+ T cells, however, in method section it is mentioned to be quantified as H-score. Besides, please provide the levels of exhausted T cell and regulatory T cell fractions in addition to CD8+ or CD4+ T cells, as their presence in the tumor is key to determine the immunosuppressive microenvironment.”

     Authors' response – The control image in Fig 9C was replaced with a better one. We thank Reviewer #2 for pointing out the inconsistency in the description of CD8+ cell quantification, which was based on CD8+ % rather than the H-score. We have corrected this error (lines 140-142, marked with red). Escape from CD8+ T cells-mediated cytotoxicity can be achieved via CD8+ T cell exclusion and/or exhaustion. As the CD8+ T cell content was significantly increased in 4T-1 tumors treated with taxifolin (Figure 9C, D), our observations suggest that attenuation of CD8+ T cell exclusion was a potential mechanism contributing to taxifolin-derived suppression of tumorigenesis. While we acknowledge the importance of examining the activation status of CD8+ T cells, we recognize that it is a more complex task that would require a thorough analysis better suited for future studies.

  1. “I would recommend testing combination therapy of taxifolin together with the checkpoint inhibitor like anti-PDL1. As low immune infiltration is one of the major causes of low efficacy of immunotherapy in solid tumors like breast cancer. If taxifolin increases the immune infiltration in tumor as indicated, it would be interesting to see if taxifolin treatment can increase the sensitivity of breast tumors for immunotherapy in combination treatment.”

     Authors' response – We strongly agree with the Reviewer on a combinational therapy using taxifolin and immune checkpoint blockade! But we thought it might be proper to leave it for future studies. In this regard, we are planning for a clinical trial to examine this potential.

Minor comments

  1. “Many grammatical and typo errors were found throughout manuscript. Please correct the spelling of ‘suppressor’ in the title of the manuscript.”

     Authors' response – We apologize for the error and have corrected it. Furthermore, we have thoroughly checked grammatical and spelling errors, and trust this revision is a better one.

  1. “Please provide scale bar for Fig 2D and S3C IHC images or indicate the magnification in figure legend.”

     Authors' response – Data presented in Fig 2D and S3C were derived from the Human Protein Atlas (https://www.proteinatlas.org/) website. Scale bars were not available.

  1. “Error bars need to be defined together with a measure of central tendency for all the box plots used in the manuscript figures”.

     Authors' response – We have ensured adding all error bars to box plots.

  1. “Figure reference for given result is not matching in page 11, line 327; ‘Additionally, CSTA and KRT79 were only 326 downregulated at the protein level in BC (Figure 4B)’. Should it be Fig 3B? Please confirm.”

     Authors' response – Thanks for pointing out the error. Yes, it should be Figure 3B. Corrected it (line 339, marked with red).

  1. “Please mention how statistical analysis was performed for Fig 8 in figure legend.”

          Authors' response – the information has been included (lines 477-478, marked with red).

“The manuscript require little bit reframing of sentences for better presentation. However, overall quality is good and acceptable. Few places have grammatical and typo-errors that needs to be corrected”.

     Authors' response – We have made a thorough effort to correct grammatical errors and typos.

Reviewer 3 Report

Following are my comments for the manuscript:

1) Nice classification of different types of breast cancers in introduction; there is a typo/spelling error in line 67 "Latter" should be 'later'; same typo in line 90

2) Impressive description of taxifolin in various cancer subtypes including its potential mechanism of actions

3) Line 196 is confusing with respect to dosage; is it mg/g or mg/kg? Please clarify

4) Statements on lines 197-198 contradicts with figure 1 A and 1 B and its figure legends as text says mg/kg as dosage and figure says uM as dosage; please justify and clarify the discrepancies 

5) Please explain RNA seq data in more details for figure 1: How were RNA extracted from tumours? More interestingly from the samples that were treated with taxifolin as there would be differences in tumour sizes at the points of RNA extraction between treated and untreated groups 

6) For figure 1 C and D: Would be interested to see entire tumour progression curves; C and D shows tumours at Day 20 only; would be interesting to see if there was prolonged tumour control upon taxifolin treatment or was there a complete regression of tumours or did the tumour relapsed? 

7) Impressive description of DEGs in table 1; nice description and evidence of 4 specific genes in tumour suppression in lines 303-316

8) In line 327, the description cites figure 4B but actually it should be figure 3B; correct the typo

8) Impressive work in figure 3 through 6 especially with nice description of DEG36 as BC mortality predictor especially in basal BC

9) Nice correlation of sigDEG23 with DEG36 in figure 7 with respect to different subtypes; however, figure 7A legends are missing the description of which colour indicates which group; it would be good to clarify

10) Validation of DEG36 and sigDEG23 using TCGA atlas in figure 8 is impressive

11) For figure 9: Have you considered quantification of TILs-CD8 cells using flow upon tumour digestion? I would be interested to see the phenotype of TIL CD8 T cells upon taxifolin treatment

12) For figure 9: At which time point the IHCs post taxifolin treatment were conducted? Is higher % of CD8+ T cells in treatment group an attribute of differences in tumour sizes?  

13) Based off discussion points made in line 502-508, any thoughts on measurement of ROS post taxifolin treatment in cells? It could be post in vitro or in vivo treatment. 

14) Any comments on checking metabolic fitness of cells post treatment by using seahorse/OCR?

15) Discussion lines 529-536 are bit vague with respect to HRNR, FLG2 and KP2P levels; any comments on discrepancy in genes at mRNA vs protein levels?

16) Typo/spelling error in line 555 "latter"

There are tons of language errors throughout the manuscript; especially spelling errors:

1) Line 26-28 has grammar errors

2) Line 136 has grammar errors in method section description

3) Line 198 has grammar errors and line 212 has typo (ideally it should be p<0.05

Author Response

We thank the reviewer for his/her overall positive comments and insightful remarks. Here are our detailed revisions.

1) “Nice classification of different types of breast cancers in introduction; there is a typo/spelling error in line 67 "Latter" should be 'later'; same typo in line 90”

16) “Typo/spelling error in line 555 "latter"”

Authors' response – Lines 67 and 90 remain unchanged in this revision. Line 555 in the last submission is line 585 in this revision. We have consulted our colleagues; the consensus is “latter” being a better choice here.

2) “Impressive description of taxifolin in various cancer subtypes including its potential mechanism of actions”

Authors' response – We appreciate Reviewer #3’s positive comment.

3) “Line 196 is confusing with respect to dosage; is it mg/g or mg/kg? Please clarify”

Authors' response – We apologize for these errors. It should be mg/kg; all mg/g presentations have been changed to mg/kg (line 200, marked with red; also see Figure S1 legend).

4) “Statements on lines 197-198 contradicts with figure 1 A and 1 B and its figure legends as text says mg/kg as dosage and figure says uM as dosage; please justify and clarify the discrepancies”

Authors' response – In this study, cells were treated with taxifolin based on µM doses, while mice were treated with taxifolin according to the mg/kg dose setting. Figure 1A and 1B present data derived from 4T-1 cells in vitro involving different µM doses of taxifolin. We hope these explanations have clarified the confusion.

5) “Please explain RNA seq data in more details for figure 1: How were RNA extracted from tumours? More interestingly from the samples that were treated with taxifolin as there would be differences in tumour sizes at the points of RNA extraction between treated and untreated groups.”

Authors' response – We thank the reviewer for these insightful comments. Detailed methodologies used for RNA sequencing and data analysis have been included (lines 163-175, marked with red). An equal amount of RNA per tumor was used for RNA seq analysis (line 166-167, marked with red), which normalized the differences in tumor volume, particularly for tumors treated with taxifolin. Additional normalization was achieved during analysis of RNA sequencing read using Galaxy.

6) “For figure 1 C and D: Would be interested to see entire tumour progression curves; C and D shows tumours at Day 20 only; would be interesting to see if there was prolonged tumour control upon taxifolin treatment or was there a complete regression of tumours or did the tumour relapsed?”

Authors' response – We appreciate these insightful remarks. Similar comments were also raised by Reviewers #1 and #2. A new panel (Figure 1D) was arranged to show the profiles of tumor growth in the DMSO- and Tax-treated groups. The corresponding Figure 1D in the last submission is Figure 1E in this revision; this panel presents the time frame of reaching endpoint, which was defined as a decline in health and/or tumor size; this figure panel reflects tumor progression.

With respect to tumor regression or relapse, we agree this being an important aspect for taxifolin-derived anti-tumor actions. However, we thought it might be more proper to systemically analyze this task in future.

For our detailed response to similar remarks raised by other two reviewers, please see the following.

     “A new panel (Figure 1D) was arranged to show the profiles of tumor growth in the DMSO- and Tax-treated groups. We noticed a wide range of variations in the growth of 4T-1 syngeneic tumors, which was in part due to distant metastasis. It is important to note that in the 4T-1 model, primary tumors often exhibit slow growth accompanied by concurrent distant metastasis. In this regard, the growth profile of primary 4T-1 tumors may not reveal the impact of taxifolin on tumorigenesis. This is a concern as 4T-1 tumors in the DMSO group were more likely to metastasize to other sites compared to those in the Tax group. We acknowledge this limitation, as addressing the metastasis issue would require a large number of animals and a more optimized experimental design. To overcome this limitation in the current experimental setting and provide a more comprehensive assessment, we performed a median-based analysis of tumor volume at a particular time point (Figure 1C) and survival analysis (Figure 1D in the last submission and Figure 1E in this revision). These messages are included in this revision (lines 208-218, marked with red). These additional analyses, combined with the tumor growth profile (Figure 1D this revision) and survival curve (Figure 1C and Figure 1E), allow for a more precise evaluation of the impact of Tax on 4T-1 tumorigenesis.”  

7) “Impressive description of DEGs in table 1; nice description and evidence of 4 specific genes in tumour suppression in lines 303-316”

Authors' response – We appreciate Reviewer #3’s positive comment.

8) “In line 327, the description cites figure 4B but actually it should be figure 3B; correct the typo”

Authors' response – Thanks for pointing out the error. Yes, it should be Figure 3B. Corrected it (line 339, marked with red).

8) “Impressive work in figure 3 through 6 especially with nice description of DEG36 as BC mortality predictor especially in basal BC”

Authors' response – We appreciate Reviewer #3’s positive comment.

9) “Nice correlation of sigDEG23 with DEG36 in figure 7 with respect to different subtypes; however, figure 7A legends are missing the description of which colour indicates which group; it would be good to clarify”

Authors' response – Color indicators for the sigDEG23 with DEG36 graph bars have been included (line 429, marked with red).

10) “Validation of DEG36 and sigDEG23 using TCGA atlas in figure 8 is impressive”

Authors' response – We appreciate Reviewer #3’s positive comment.

11) “For figure 9: Have you considered quantification of TILs-CD8 cells using flow upon tumour digestion? I would be interested to see the phenotype of TIL CD8 T cells upon taxifolin treatment”

Authors' response – We quantified TILs-CD8+ T cells using IHC. This study was conducted during the challenging period of the pandemic, during which quantification of these cells by flow has been surprisingly challenging. Nonetheless, we may do this in our following research.

12) “For figure 9: At which time point the IHCs post taxifolin treatment were conducted? Is higher % of CD8+ T cells in treatment group an attribute of differences in tumour sizes?”

Authors' response – At the endpoint. As tumors treated with taxifolin reached endpoint in a significantly delayed manner (see Figure 1E), this time point used to quantify TILs-CD8+ T cells was thus justified.

13) “Based off discussion points made in line 502-508, any thoughts on measurement of ROS post taxifolin treatment in cells? It could be post in vitro or in vivo treatment.”

14) “Any comments on checking metabolic fitness of cells post treatment by using seahorse/OCR?”

Authors' response – Given cancer cells are commonly associated with metabolic alterations, these remarks are highly insightful, to which we thank the reviewer for these comments. We have briefly discussed this aspect with references included (lines 561-572, marked with red).

15) “Discussion lines 529-536 are bit vague with respect to HRNR, FLG2 and KP2P levels; any comments on discrepancy in genes at mRNA vs protein levels?”

Authors' response – We share the Reviewer’s insights; the lack of increases of mRNA expression for these genes in BCs with increases in their gene copies is intriguing; the underlying mechanisms should be investigated in future. This research may have great potential in BC therapy. In this revision, we have provided some thoughts on this issue (lines 539-544, marked with red). We hope that these additional considerations contribute to the overall discussion and help stimulate further research in this area.

“There are tons of language errors throughout the manuscript; especially spelling errors:”

Authors' response – We apologize for the errors and have thoroughly checked grammatical and spelling errors. We trust this revision is a better one.

1) “Line 26-28 has grammar errors”

Authors' response – Corrected it.

2) “Line 136 has grammar errors in method section description”

Authors' response – Corrected it (lines 140-142, marked with red).

3) “Line 198 has grammar errors and line 212 has typo (ideally it should be p<0.05”

Authors' response – Corrected it. With respect to q < 0.05 in line 212 in the last submission and line 222 in this submission, q <0.05 is more stringent than p < 0.05, i.e. when q < 0.05, p will be < 0.05. In this regard, we think q <0.05 can be used. We hope that Reviewer #3 will agree.

Round 2

Reviewer 1 Report

The author's revisions are addressed all y concerns.